# The effect of DNA polymorphisms and natural variation on crossover hotspot activity in Arabidopsis hybrids

Maja Szymanska-Lejman [1], Wojciech Dziegielewski [1], Julia Dluzewska[1], Nadia Kbiri[1], Anna Bieluszewska [1], R. Scott Poethig[2] & Piotr A. Ziolkowski [1] ✉

In hybrid organisms, genetically divergent homologous chromosomes pair and recombine during meiosis; however, the effect of specific types of polymorphisms on crossover is poorly understood. Here, to analyze this in Arabidopsis, we develop the seed-typing method that enables the massively parallel fine-mapping of crossovers by sequencing. We show that structural variants, observed in one of the generated intervals, do not change crossover frequency unless they are located directly within crossover hotspots. Both natural and Cas9-induced deletions result in lower hotspot activity but are not compensated by increases in immediately adjacent hotspots. To examine the effect of single nucleotide polymorphisms on crossover formation, we analyze hotspot activity in mismatch detection-deficient *msh2* mutants. Surprisingly, polymorphic hotspots show reduced activity in *msh2*. In lines where only the hotspot-containing interval is heterozygous, crossover numbers increase above those in the inbred (homozygous). We conclude that MSH2 shapes crossover distribution by stimulating hotspot activity at polymorphic regions.

Crossover (CO) recombination plays a key mechanical role during the first meiotic division by ensuring proper chromosome segregation[1,2]. Together with the independent assortment of homologous chromosomes, COs are also the main source of genetic diversity in the population and are largely responsible for shaping adaptability to a changing environment[3–5]. In this context, the relationship between genetic diversity and meiotic recombination is particularly important. Despite many years of research, how CO distribution is shaped by DNA interhomolog polymorphisms remains poorly understood.

Meiotic recombination starts with the formation of programmed DNA double-strand breaks (DSBs)[6]. Following end resection, one DNA strand invades the homolog and is extended by DNA polymerase, forming a displacement loop. Multiple rounds of strand invasion, extension, and displacement often occur at this stage and create complex recombination intermediates[7,8]. Many of these intermediates are dissociated by DNA helicases and are repaired via synthesis-dependent strand annealing (SDSA). SDSA

invariably leads to a noncrossover (NCO), where only a very short patch of DNA has been modified. Alternatively, second-end capture can form a double Holiday junction (dHJ), which is resolved via recombination pathways involving nucleases leading to either COs or NCOs[2,9,10].

Usually, only a small fraction of meiotic DSBs is repaired by COs (5–10% in *Arabidopsis thaliana*), with most being repaired as NCOs. At least two recombination pathways can lead to crossover in most eukaryotes, including plants. The first pathway, called the ZMM pathway, is responsible for the formation of about 85% of all COs, which are referred to as Class I COs. Class I COs exhibit genetic interference, whereby the distance between two CO events on the same chromosome is greater than that expected from a random distribution. The frequency and placement of Class I COs are controlled by the abundance of the pro-crossover factor HEI10 and the synaptonemal complex (SC), a tripartite structure that forms between paired homologous chromosomes in prophase I[11–16]. The remaining crossovers, so-called

[1]Laboratory of Genome Biology, Institute of Molecular Biology and Biotechnology, Adam Mickiewicz University, Poznań, Poland. [2]Department of Biology, University of Pennsylvania, Philadelphia, PA 19104, USA. ✉e-mail: pzio@amu.edu.pl

Class II COs, show no interference and are largely dependent on the structure-specific nuclease MUS81[17,18].

COs usually occur in narrow chromosomal regions called CO hotspots[19–23]. There are few studies on hotspots in plants, and new technical approaches would bring the field forward[24]. However, plant hotspots are known to be primarily located in gene promoters and terminators and are associated with active chromatin modifications, including low nucleosome density, the histone variant H2A.Z, tri-methylation at lysine 4 of histone H3 (H3K4me3), and low DNA methylation[20,25]. In line with these observations, CO hotpots are much less common in pericentromeric regions than along chromosome arms[26]. Importantly, pericentromeres tend to show a much higher level of polymorphisms, as manifested by both the presence of structural variants and a higher density of single nucleotide polymorphisms (SNPs)[27–29].

Structural variation, including insertions, deletions, and loss of similarity, have been associated with local suppression of COs[30–32]. Sequence polymorphisms between homologous chromosomes may also result in base pair mismatches within recombination inter-mediates, which affects recombination outcomes[33]. The detection of such mismatches during meiotic recombination is often dependent on MutS-related heterodimers that contain MSH2[34–39]. MSH2 hetero-dimers bind mismatches to prevent post-replicative mutations, while they often trigger heteroduplex rejection during recombination[40–43]. Interestingly, while the mitotic functions of MSH2 in heteroduplex rejection and correction of mismatches in heteroduplex DNA are conserved across eukaryotes, the role(s) of MSH2 in meiotic recom-bination may be different. For example, mismatches within recombi-nation hotspots result in strand rejection and NCO repair in budding yeast[34,35]. However, loss of MSH2 function does not substantially change the meiotic CO frequency in mouse hotspots in crosses between strains exhibiting 0.6 and 1.8% sequence polymorphisms[44]. In Arabidopsis hybrids, CO midpoints are positively associated with interhomolog SNPs at the kilobase (kb) scale, and this association is significantly weakened in the *msh2* background[39]. These observations suggest that MSH2 heterodimers may exert different functions in meiotic recombination. At a genomic scale, a positive correlation was observed between historical recombination as measured via linkage disequilibrium and sequence diversity in many species[45–49]. In addition, the pattern of heterozygosity along the chromosome determines CO distribution. In such a genetic context, wild-type Arabidopsis shifts COs from homozygous to heterozygous regions on a chromosome. This homozygosity–heterozygosity juxtaposition effect is interference dependent and disappears in the *msh2* mutant background[39,50]. On the other hand, it has recently been reported that the megabase-scale crossover landscape is largely independent of sequence divergence, with recombination distributions in Arabidopsis pure lines being very similar to those of intra-specific hybrids[51]. This supports the notion that much of the association between CO rates and SNPs is due to the mutagenic effects of recombination rather than a pro-crossover influence of SNPs[51].

Here, we develop the seed-typing method to study Arabidopsis crossovers at the hotspot scale. Seed-typing relies on the selection and targeted sequencing of recombinants within short genetic intervals. One of the intervals we developed, called Chili Pepper (ChP), includes three well-separated recombination hotspots. We observe that local structural polymorphisms have limited impact on a crossover, though deletions within the strongest hotspot cause a dramatic drop in CO rates. Based on seed-typing, we do not find evidence for a short-distance hotspot competition. Within a hotspot, crossovers tend to occur in regions devoid of polymorphism. However, we observe that hotspots with higher overall SNP levels show a reduction in CO activity in the *msh2* mutant when compared to the wild type. In recombinant lines where only the ChP interval is heterozygous while the remainder of the genome is homozygous, ChP CO frequency is significantly higher

than in fully homozygous inbreds. We do not observe these results in the *msh2* mutant. We propose that MSH2 complexes detecting local polymorphisms stimulate Class I COs at the hotspot scale.

## Results

### Construction of extremely short interval lines applicable to seed-typing

We developed a system for precise CO mapping that enables robust analysis of recombination at the hotspot scale. This method is based on extremely short interval lines (ESILs), which carry fluorescent reporters expressed in seeds, separated by no more than 50 kb. The fluorescently tagged intervals provide both precise CO frequency measurements[11,50,52–56] and an easy assay for recombinants based on seed fluorescence. We used a recently developed large-scale collection of fluorescent reporter lines in Arabidopsis[57] to construct five ESILs by crossing the selected single-color reporter lines in the Col-0 back-ground (Supplementary Fig. 1). The intervals are located on the upper arm of chromosome 3 near the telomere (BB), interstitially forming adjacent intervals (Bow Tie [BT], TG, and BF), or in the pericentromeric region (ChP) (Fig. 1a and Supplementary Table 1).

To assess the applicability of the ESILs for fine-scale CO analysis, we used a published ultra-high-density CO map created for a cross between the Arabidopsis Col-0 and L*er*−0 accessions[26]. To this end, we plotted the CO map for chromosome 3 with a window size of 20 kb and superimposed the location of the intervals from the five ESILs (Fig. 1a). Notably, the ChP interval coincided with one of the three most highly recombining sites on chromosome 3 out of 1169 windows (Fig. 1a). Next, we crossed the ESILs to the Arabidopsis accessions Col-0, Ct-1, and L*er*−0 (hereafter Col, Ct, and L*er*, respectively) to measure the CO frequency in the nearly isogenic lines, whose parents differ only in the presence of fluorescent reporters (for the sake of simplicity, we will call them inbreds) and in the hybrids, whose parents differ along the entire genome (Fig. 1b and Supplementary Table 1). In inbreds, all intervals showed a mean CO frequency of 3.5 cM/Mb or greater. The exception was TG, with a very low CO frequency. By contrast, the pericentromeric ChP interval exhibited the highest CO frequency (mean of 9.09 cM/Mb) (Fig. 1b). Measurements in hybrid contexts returned values similar to those observed in their respective inbreds, which was consistent with previous observations that the presence of polymorphisms does not inhibit COs (Fig. 1b)[39,50,51]. Again, the most interesting interval was ChP, which had a substantially higher CO frequency in hybrids (18.22 cM/Mb for Col-ChP × L*er* and 14.52 cM/Mb for Col-ChP × Ct) relative to the genome average (3.2 cM/Mb for Col × L*er*[58]).

### Determination of CO hotspots in the highly polymorphic ChP interval

Due to its pericentromeric location, the ChP interval exhibited the highest density of Col × L*er* interhomolog polymorphisms (18.7 poly-morphisms/kb; Supplementary Table 2). Together with its very high CO frequency, this high degree of polymorphisms makes ChP an excellent system for the high-resolution mapping of CO events. To determine the positions of potential recombination hotspots within the ChP interval, we developed the seed-typing method, where F$_2$ recombinant seeds from hybrid plants are selected based on the expression of a single fluorescent reporter indicating a CO event (Fig. 1c, d). Genomic DNA is isolated from seedlings obtained from preselected recombinants and used to amplify the entire interval by high-fidelity long-range PCR (LR-PCR; Fig. 1c) using three overlapping amplicons. We pooled the LR-PCR products for each recombinant and used them for library preparation using unique barcodes. We then pooled the libraries and sequenced them to a depth of ~1500×. For 81.5% recombinants (243 out of 298), we successfully and precisely determined the location of COs, defined as a transition between parental/heterozygous genotypes, based on the SNPs distinguishing Col and L*er* sequences (Fig. 1e and Supplementary

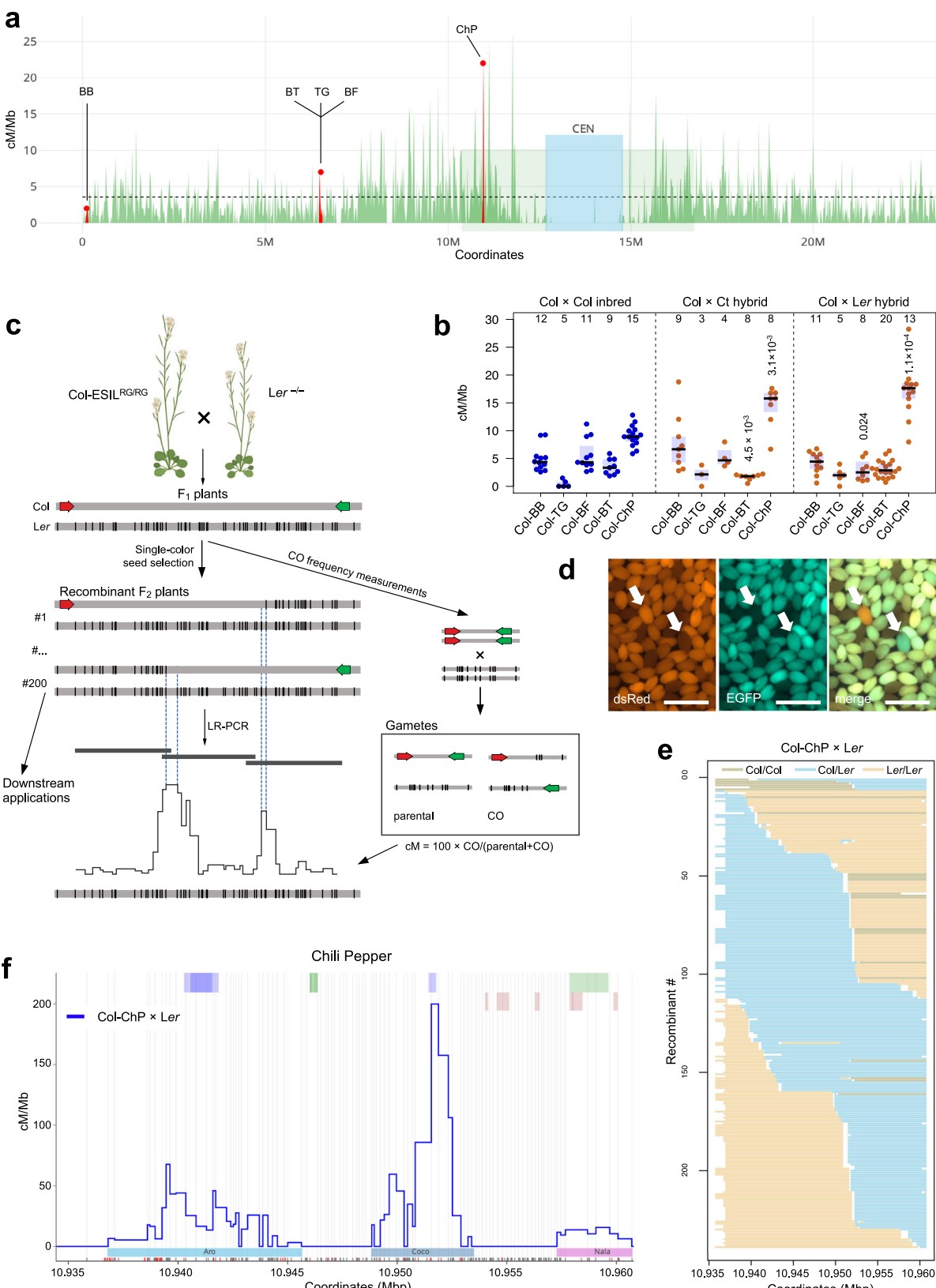

Fig. 2). By superimposing the location of all identified COs, we obtained a high-resolution map for ChP (Fig. 1f).

We identified 242 COs that are unevenly distributed across the 26.3-kb region, revealing the presence of three clearly separated hotspots, which we named Aro (8.8 kb), Coco (4.6 kb), and Nala (3.4 kb) (Fig. 1e, f and Supplementary Table 3). These hotspots were located within three of the four short genes present in ChP and differed in their recombination frequency: Aro accounted for 32% of the identified CO events, 60% for Coco, and only 8% for Nala. The activity of each hotspot can be determined by relating these values to the measurement of the CO frequency for the entire interval, which is performed by seed scoring (Fig. 1b, c). Each hotspot showed a CO frequency substantially

**Fig. 1 | CO analysis with seed-typing. a** Col × L*er* CO map[26] for chromosome 3 with the location of the five ESILs marked in red. CO frequency (green) was plotted in 20-kb windows. The positions of pericentromeres and the genetically defined centromere are shown with light-green and blue rectangles, respectively. **b** CO frequency in the five ESILs, as measured by fluorescent seed scoring in inbreds (blue) and hybrids (red). The center line of a boxplot indicates the mean; the bounds indicate the 75th and 25th percentiles. Each dot represents measurements from one individual. The number of individuals are indicated above the boxplots. Two-sided *P* values indicate the statistical difference in CO rates in hybrids versus inbreds (Welch's *t*-test). **c** Diagram of seed-typing. Crossing the Col-ESIL to a different accession (e.g., L*er*) results in F1 hybrids. Segregation of fluorescent reporters in seeds obtained from F1 plants enables both CO frequency measurements in the interval (right panel) and the selection of single-color recombinant seeds, which are grown to isolate genomic DNA. Libraries for sequencing from all recombinants are prepared based on the amplicons obtained after LR-PCR. The CO landscape is

determined by CO sites identified based on SNPs. **d** Selection of recombinants from Col-ChP × L*er* F2 seeds based on fluorescence. Scale bars, 2 mm. Arrows indicate the location of --/-R (top) and G-/-- (bottom) seeds corresponding to L*er*/L*er*→Col/L*er* and Col/L*er*→L*er*/L*er* genotypes, respectively. **e** Genotypes of 243 Col-ChP × L*er* recombinants. Each horizontal line corresponds to one recombinant with color-encoded genotype. The x-axis indicates coordinates within ChP. **f** CO landscape within ChP for a Col-ChP × L*er* cross obtained by seed-typing. COs (blue) are normalized to the ChP CO frequency measured by seed scoring. SNPs spaced at least 100 bp apart, shown as vertical gray lines, were used to determine the CO topology. Black and red x-axis ticks correspond to all Col/L*er* SNPs and InDels, respectively. Genes are shown as light-green (forward) and dark-green (reverse) rectangles, and transposons are shown as blue rectangles. Positions of three CO hotspots are indicated in colored rectangles below the plot. Source data are provided as a Source Data file.

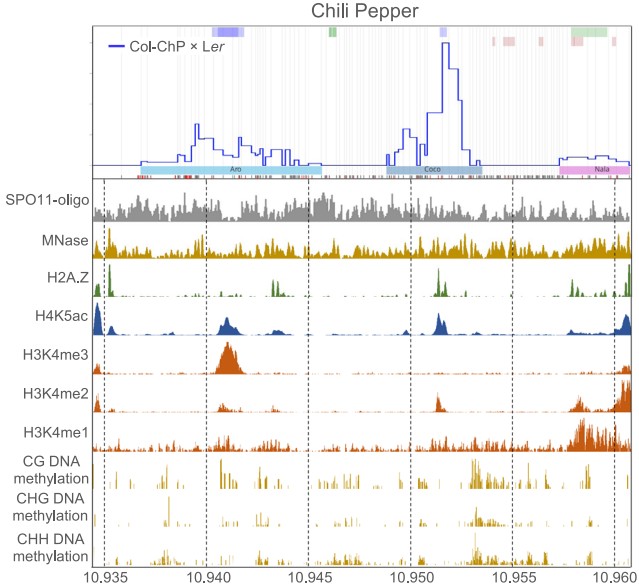

**Fig. 2 | Chromatin landscape in the ChP interval.** Histograms showing normalized coverage values for SPO11-1-oligonucleotides[85], nucleosome occupancy (MNase-seq)[86], H2A.Z[86], H4K5ac[78], H3K4me1/me3[87], H3K4me2[88], and DNA methylation (BSseq) in CG, CHG, and CHH sequence contexts[89] were presented in relation to ChP CO landscape for Col-ChP × L*er*. Source data are provided as a Source Data file.

higher than the genome average, with Coco reaching up to 62.14 cM/Mb (Supplementary Table 3).

Although ChP is highly polymorphic, COs occurred at places with relatively few polymorphisms. Hotspots did not show an increased number of DSBs relative to the rest of the interval (Fig. 2). We analyzed the profiles of various chromatin modifications over the ChP interval and observed that recombination hotspots are associated with an open chromatin structure, high histone acetylation, and the presence of H2A.Z (Fig. 2). Notably, the Coco hotspot showed high levels of H3K4me2, but not H3K4me3 (Fig. 2).

### Influence of structural polymorphisms on CO frequency within ChP

To assess the CO rate and enable recombinant selection, the eGFP and dsRed 3.2-kb reporter cassettes in the Col-ChP line are required in the hemizygous state. Genetically, in this state, they can be considered kilobase insertions. Since ChP is only 26.3 kb, the CO frequency in this interval may potentially be influenced by the presence of these two insertions. To test this possibility, we tried to construct a pseudo-reporter line in which the reporter cassettes were preserved but produced nonfunctional fluorescent proteins. Crossing the Col-ChP line to

such a Col-pseudo-ChP (Col-ΨChP) line would allow the measurement of recombination (with the Col-ChP fluorescent reporters) and quantify the effect of the T-DNA insertion when compared to Col-ChP × Col cross. To generate a Col-ΨChP line, we simultaneously targeted the dsRed and eGFP coding sequences by Cas9-mediated gene editing in the Col-ChP background and selected nonfluorescent seeds in successive generations of transformants having segregated out the CRISPR-Cas9 construct (Fig. 3a). In most cases, the loss of fluorescence was due to cassette silencing, which made them unsuitable for further experiments. However, we obtained one line in which the dsRed cassette carried a frameshift deletion while the eGFP cassette was largely removed (Supplementary Fig. 3). The cross of this Col-ΨChP line with Col-ChP only showed a slight decrease in ChP recombination (Fig. 3b), indicating that the high CO frequency in ChP was not due to the insertion of the dsRed cassette. Moreover, the ChP CO measurements for Col-ChP × L*er* are consistent with the recombination estimates for this interval according to the Col × L*er* F2 map[26] (18.2 cM/Mb and 22.7 cM/Mb respectively; see also Fig. 1a, b). We concluded that DNA insertions have a limited effect on recombination within the interval.

The chromosomal region containing the ChP interval is structurally very polymorphic between different Arabidopsis accessions. To investigate how these differences might affect recombination frequency in ChP, we used accessions whose genomes were recently resequenced with high-quality chromosome-level assemblies[59]. Two of these accessions were structurally similar to Col over the ChP region, while four other accessions had a long highly diverged region of more than 7 kb located upstream of ChP (Fig. 3c and Supplementary Fig. 4). Moreover, the L*er* accession also carried a large 8.7-kb diverged region downstream from the interval (Fig. 3c and Supplementary Fig. 4). Except for C24, all other hybrids obtained from a cross with Col-ChP had a significantly higher CO frequency than their corresponding inbred (Fig. 3d). Surprisingly, we noticed that large structural changes located on one (Sha, Cvi, and Kyo) or both (L*er*) sides of the ChP interval did not result in a significantly different ChP CO frequency compared to accessions lacking these changes (An, Eri) (Welch's *t*-test values indicated on Fig. 3d). Collectively, the data from crosses involving the pseudoreporter line and structurally diverged accessions show that even large structural alterations involving several kilobases do not have a significant effect on COs within ChP if they do not overlap with CO hotspots.

### CO hotspots are highly conserved between Arabidopsis diverged accessions and show no competition

The presence of three distinct CO hotspots within the ChP interval makes it an ideal model for testing competition between hotspots and the conservation of the recombination landscape between Arabidopsis accessions. From the crosses analyzed by seed scoring, only Col-ChP × C24 showed a CO frequency that was not statistically different from the inbreds (mean 8.73 vs. 8.54, respectively, Fig. 3d).

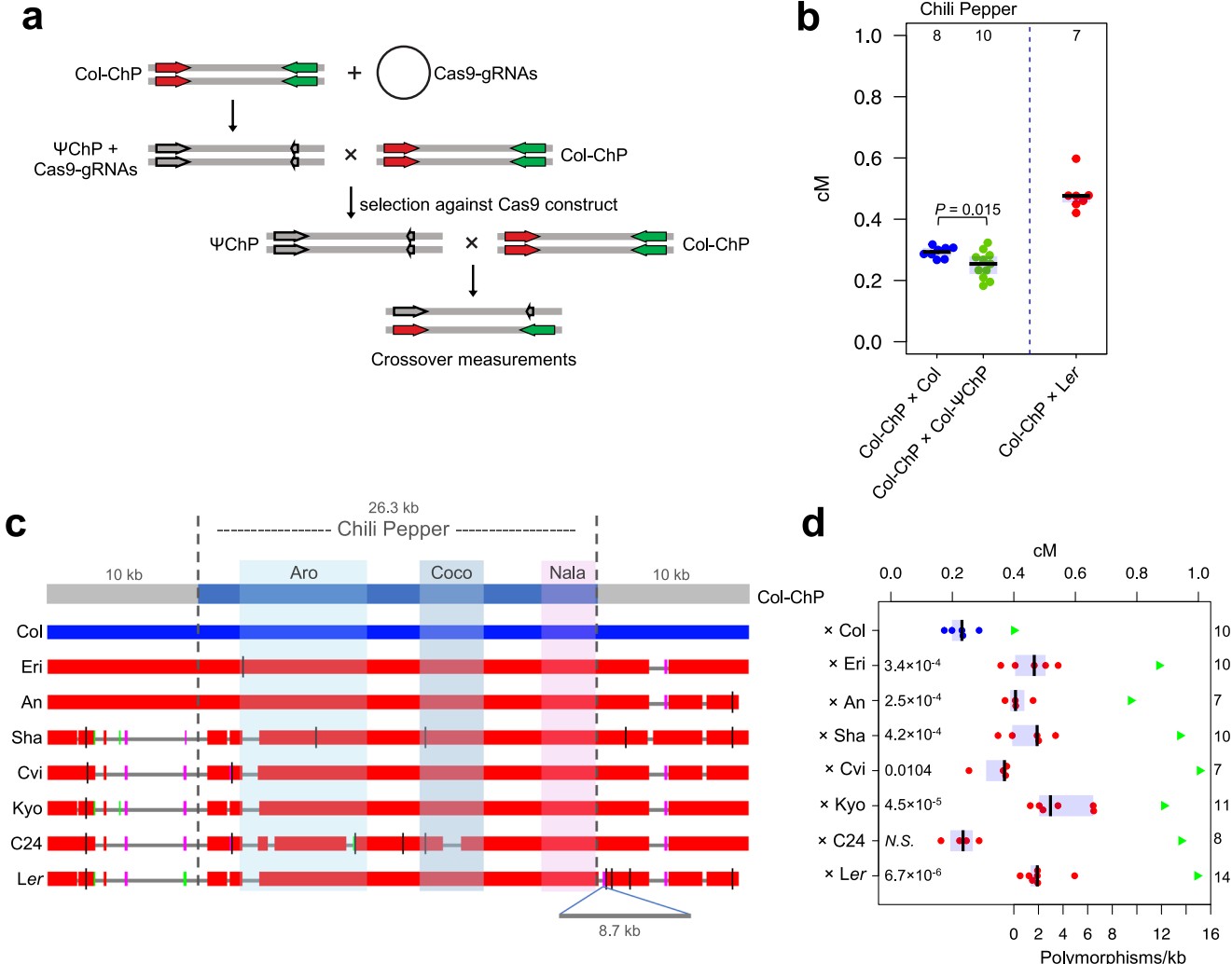

**Fig. 3 | Effects of InDels and structural variation on CO frequency within the ChP interval. a** Generation of Col-pseudo-ChP (Col-ΨChP). Cas9 sgRNAs were used to simultaneously generate mutations in the dsRed and eGFP cassettes, leading to the loss of fluorescence. Nonfluorescent seeds were sequenced and crossed with Col-ChP to measure CO frequency. **b** CO frequency for Col × Col-ChP and Col × Col-ΨChP. Measurements for Col-ChP × L*er* are shown for comparison. The center line of a boxplot indicates the mean; the upper and lower bounds indicate the 75th and 25th percentiles, respectively. Each dot represents a measurement from one individual. The number of individuals are also indicated above the boxplots. The two-sided *P* value was estimated by Welch's *t*-test. **c** Alignments of the ChP region between different Arabidopsis accessions. The blue horizontal bar shows the query sequence (Col), red horizontal bars indicate homologous sequences in other accessions, colored vertical tick marks indicate short regions of lower similarity, and gray lines indicate nonhomologous regions. Insertion of an 8.7-kb fragment next to the ChP interval in L*er* is also indicated. **d** CO frequency (in cM) for different crosses between Col-ChP and the accessions shown in (**c**). The center line of a boxplot indicates the mean; the upper and lower bounds indicate the 75th and 25th percentiles, respectively. Each dot represents a measurement from one or two pooled individuals. The numbers of individuals are indicated on the right from the boxplots. Two-sided *P* values indicate the statistical difference in ChP CO rate in hybrids versus inbreds (Welch's *t*-test). Green triangles indicate the polymorphism level (SNPs+InDels) for each cross within the ChP interval. Source data are provided as a Source Data file.

Sequence analysis showed that C24 carries a deletion of 1194 bp at the position overlapping the Coco hotspot (Fig. 3c). Using Col-ChP × C24 seed-typing, we measured a significant drop in the number of CO events within Coco (Fig. 4a, b and Supplementary Fig. 5). Although we observed a dramatic decrease in the CO number in the immediate vicinity of the deletion, we detected recombination events in the genomic neighborhood, indicating that the Coco hotspot still retains some of its activity. A CO landscape comparison between Col-ChP × C24 and Col-ChP × L*er* revealed a nearly identical location for all three hotspots (Fig. 4a, b and Supplementary Fig. 5). Importantly, the CO activity of Aro and Nala, the two hotspots located on either side of Coco, did not increase due to lower Coco activity (Fig. 4c). This observation suggests that there is no competition between immediately adjacent hotspots; however, this conclusion may be obscured by variability in recombination between L*er* and C24, e.g., the

existence of trans-acting CO modifiers. To eliminate this possibility, we created de novo deletions within the Coco region in the L*er* accession using a pair of single-guide RNAs (sgRNAs) to delete a central, hyperactive part of Coco. We obtained two independent lines with deletions of 714 bp (L*er*Δ#24) and 825 bp (L*er*Δ#76), respectively, which we then crossed to the Col-ChP line (Fig. 4d and Supplementary Fig. 6). Both crosses showed an approximately two-fold decrease in CO frequency compared to the Col-ChP × L*er* control (Fig. 4e). Seed-typing of Col-ChP × L*er*Δ#24 revealed a dramatic reduction in CO events in the region neighboring the deletion, while we observed no other changes in CO topology (Fig. 4f, g). In addition, a calculation of hotspot activity showed no alteration in hotspot usage (Fig. 4h). This result confirms that Arabidopsis recombination hotspots operate independently of each other and that there is no short-distance hotspot competition.

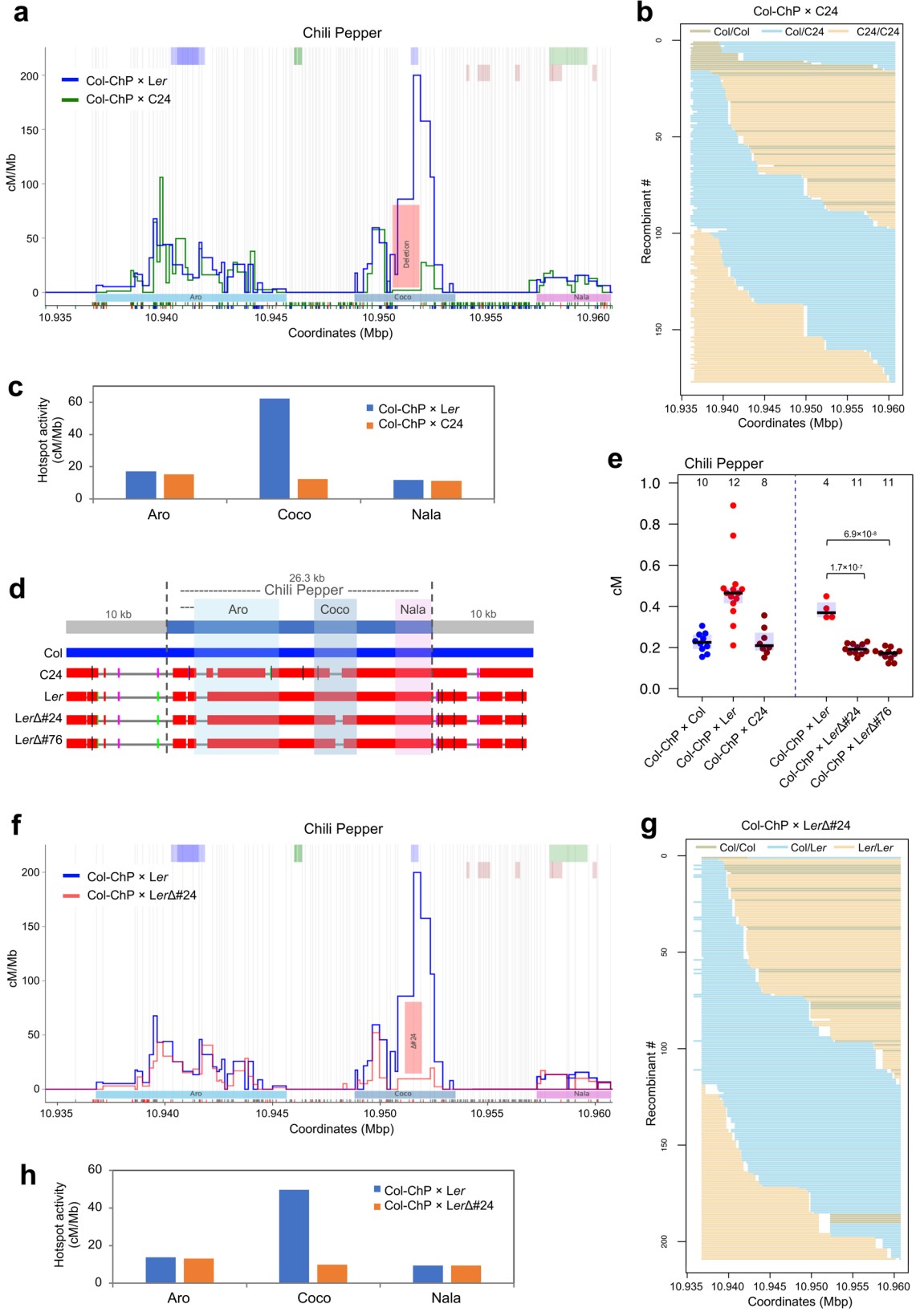

**Loss of MSH2 function results in an increase or decrease of the CO rate depending on chromosomal location**

As SNP-associated CO remodeling depends on MSH2[39], we checked how genetically inactivating *MSH2* would affect recombination in the ChP and BT intervals, the latter being interstitial and with a lower SNP density than ChP (3.58 vs. 14.20 SNPs/kb; Fig. 1a, Supplementary Fig. 7,

and Supplementary Table 2). We thus backcrossed the *msh2-2* mutation to the Col-ChP line and measured CO frequency in the Col-ChP*^msh2^* × Col*^msh2^* inbred. CO frequency was identical to that in the wild type with functional *MSH2* (Fig. 5a). We then used Cas9-mediated gene editing to introduce a null mutation at *MSH2* in the L*er* background (Supplementary Figs. 8, 9) and obtained the Col-ChP*^msh2^* × L*er^msh2^*

**Fig. 4 | Hotspot conservation and competition within ChP. a** CO landscape within ChP for the Col-ChP × C24 (green) and Col-ChP × L*er* (blue). COs are normalized to the measured ChP CO frequency. SNPs spaced at least 100 bp apart, shown as vertical gray lines, were used to determine the CO topology. Top and bottom x-axis tick marks correspond to Col/C24 and Col/L*er* SNPs, respectively, while red tick marks correspond to InDels. Deletion within Coco is indicated by the pink rectangle. **b** Genotypes of 177 Col-ChP × C24 recombinants. Each horizontal line corresponds to one recombinant with color-encoded genotype. The x-axis indicates coordinates within ChP. **c** Hotspot activity for Col-ChP × C24 and Col-ChP × L*er*. **d** Alignments of the ChP region between Col, C24, L*er*, and L*er*Δ#24 and L*er*Δ#76 deletion lines. The blue horizontal bar shows the query sequence (Col); red horizontal bars indicate homologous sequences in other lines. Colored vertical tick marks indicate short regions of lower similarity, and gray lines indicate non-homologous regions. Insertion of the 8.7-kb fragment next to the ChP interval in L*er*,

L*er*Δ#24, and L*er*Δ#76 is not shown for simplicity. **e** CO frequency for crosses between Col-ChP and the lines indicated in **d**. The center line of a boxplot indicates the mean; the upper and lower bounds indicate the 75th and 25th percentiles, respectively. Each dot represents a measurement from one individual. The numbers of individuals are also indicted above the boxplots. The two-sided *P* values were estimated by Welch's *t*-test. **f** ChP CO landscape in Col-ChP × L*er*Δ#24 (red) and Col-ChP × L*er* (blue). COs were normalized to the measured ChP CO frequency. SNPs spaced at least 100 bp apart, shown as vertical gray lines, were used to determine the CO topology. The black and red x-axis ticks correspond to all Col/L*er* SNPs and InDels, respectively. The deletion L*er*Δ#24 is indicated by a pink rectangle. **g** Genotypes of 208 Col-ChP × L*er*Δ#24 recombinants. Each horizontal line corresponds to one recombinant with color-encoded genotype. **h** Hotspot activity for Col-ChP × L*er*Δ#24 and Col-ChP × L*er*. Source data are provided as a Source Data file.

hybrid. This *msh2* hybrid showed a significantly lower CO rate than the corresponding wild-type hybrid (Col-ChP × L*er*) but comparable to the rate observed in the inbred (Fig. 5a). Next, we investigated the BT CO rate in *msh2* inbreds and hybrids (Fig. 5b). Similar to *msh2* ChP, *msh2* BT inbreds showed no change from the wild type. However, contrary to the Col-ChP*msh2* × L*er msh2* hybrid, the Col-BT*msh2* × L*er msh2* displayed a significantly higher CO frequency than its wild-type counterpart. These results suggest that MSH2-dependent changes in recombination levels observed in ChP and BT hybrids are affected by CO redistribution at the chromosome-scale and not the local effect of polymorphism.

As the pro-CO function of MSH2 is related to the ZMM pathway, we also used a line overexpressing *HEI10* (C2 line), in which the activity of the ZMM pathway is increased[11,39,50]. We observed that *HEI10* over-expression does not cause a change in ChP CO rates, in the inbred or in the hybrid context, when compared to the wild type (Fig. 5a). However, when we repeated this experiment for the BT interval, we observed an increase in CO frequency for both the inbreds and hybrids (Fig. 5b). These results suggest that the increase in CO activity via the ZMM pathway triggered by *HEI10* overexpression is not dependent on interhomolog polymorphisms but is dependent on the chromosomal location (e.g., centromere-proximity effect).

## MSH2 stimulates crossovers in hotspots in response to local polymorphism

If the loss of MSH2 function causes a change in CO in hybrids but not in inbreds, does the activity of individual hotspots also change? To answer this question, we investigated the CO landscape in Col-ChP*msh2* × L*er msh2*. The resulting CO profile was similar to that of the wild type (Fig. 5c, d). Although we observed fewer CO events throughout the ChP interval, all three hotspots were active in the *msh2* background (Fig. 5c). However, the activity of the Aro hotspot showed the largest percentage decrease in the *msh2* background compared to the wild-type background (34.5% activity of the wild type) (Fig. 5e). In turn, the activity of Nala dropped only to 70.8% of its wild-type levels (Fig. 5e). Considering that Aro and Nala are the most and the least polymorphic hotspots in ChP, respectively (Supplementary Table 3), we conclude that MSH2-dependent CO stimulation in polymorphic regions can be observed at the scale of individual hotspots.

Next, we determined the precise CO sites for 185 and 196 recombinants from the Col-BT × L*er* and Col-BT*msh2* × L*er msh2* crosses, respectively. For this purpose, we developed seed-typing for the BT interval using five overlapping LR-PCR amplicons. The BT CO landscape was very different from that of the ChP interval, as it included many hotspots close to each other (Fig. 6a–c). Due to the much lower SNP density and high gene density, individual hotspots overlapped in SNP-SNP sections, as approximated by H2A.Z peaks, making it impossible to clearly distinguish between them (Fig. 7 and Supplementary Fig. 10). Therefore, we divided the BT interval into 13 non-overlapping sections of 2.6–5.2 kb each based on the pattern of polymorphisms and the occurrence of H2A.Z peaks (Fig. 6d and

Supplementary Table 4). We excluded three sections from the analysis because they contained no CO events. We then calculated the Col/L*er* polymorphism density and *msh2* crossover activity relative to the wild type for each section. As with ChP, we also observed a clear relationship for BT between the level of polymorphism and the change in CO activity in *msh2* (Fig. 6e). In the highly polymorphic sections 11 and 9, the *msh2* mutant showed fewer CO events than the wild type, whereas the mutant had more COs in sections containing fewer than eight SNPs/kb. Interestingly, in section 10, with only 0.6 SNP/kb, the number of CO events was more than four times higher in *msh2* than in the wild type, suggesting that without functional mismatch detection, CO events mostly occur in polymorphism-free hotspots. The relationship between the change in *msh2* recombination frequency versus the wild type and the polymorphism density was statistically significant (two-sided *P* value = $2.37 \times 10^{-4}$, Spearman *rho* = −0.912). The results for ChP and BT show that MSH2 increases the chance of CO in hotspots surrounded by more SNPs. Importantly, this effect is independent of whether there is an increase (ChP) or a decrease (BT) in CO frequency for a given chromosomal interval in the background defective in mismatch recognition (Fig. 5a, b).

Our comparison of the recombination for ChP and BT in the wild type and *msh2* was based on the analysis of $F_1$ individuals that are polymorphic along the entire genome. Consequently, CO measurements reflect both the local and chromosomal-scale effects of polymorphisms. To investigate only the direct influence of interhomolog polymorphisms on CO formation in the ChP interval, we took advantage of a unique feature of our system that allows the use of recombinants in downstream analyses via successive generations (Fig. 1c). For seed-typing, we sampled 3-week-old plants, which allows recombinants to further grow and produce offspring. Based on the identified CO sites, we selected Col-ChP*msh2* × L*er msh2* recombinants with green or red fluorescence and crossed them (Fig. 8a). From their offspring, we selected $F_2$ seeds, where a new CO event occurred between fluorescent reporters, thus bringing the two reporters onto the same chromosome, with an intervening L*er* fragment in between them. We then backcrossed these lines (3×) to the Col accession. As a result, we obtained $R^2$ (Recombinant × Recombinant) lines in which only selected fragments of ChP carried Col/L*er* polymorphisms, while the remaining genome was homozygous Col (Fig. 8b and Supplementary Tables 5, 6). Importantly, all $R^2$ lines that carried Col/L*er* polymorphisms across the entire interval ($R^2$-1; ~240 SNPs) showed ChP CO frequencies significantly higher than both ChP inbreds and hybrids (Welch's *t*-test; $P \leq 1.8 \times 10^{-2}$) (Fig. 8c). Plants heterozygous only within the Coco hotspot ($R^2$-2 lines; ≥44 SNPs) also had a significantly higher ChP CO rate than pure inbreds ($P \leq 3.6 \times 10^{-2}$). The siblings of $R^2$-1 and $R^2$-2 plants in the *msh2* background exhibited a CO rate that was not different from that of the *msh2* inbred or hybrid (Fig. 8d). This result indicates that MSH2 promotes crossovers in a single hotspot based on locally occurring interhomolog polymorphisms.

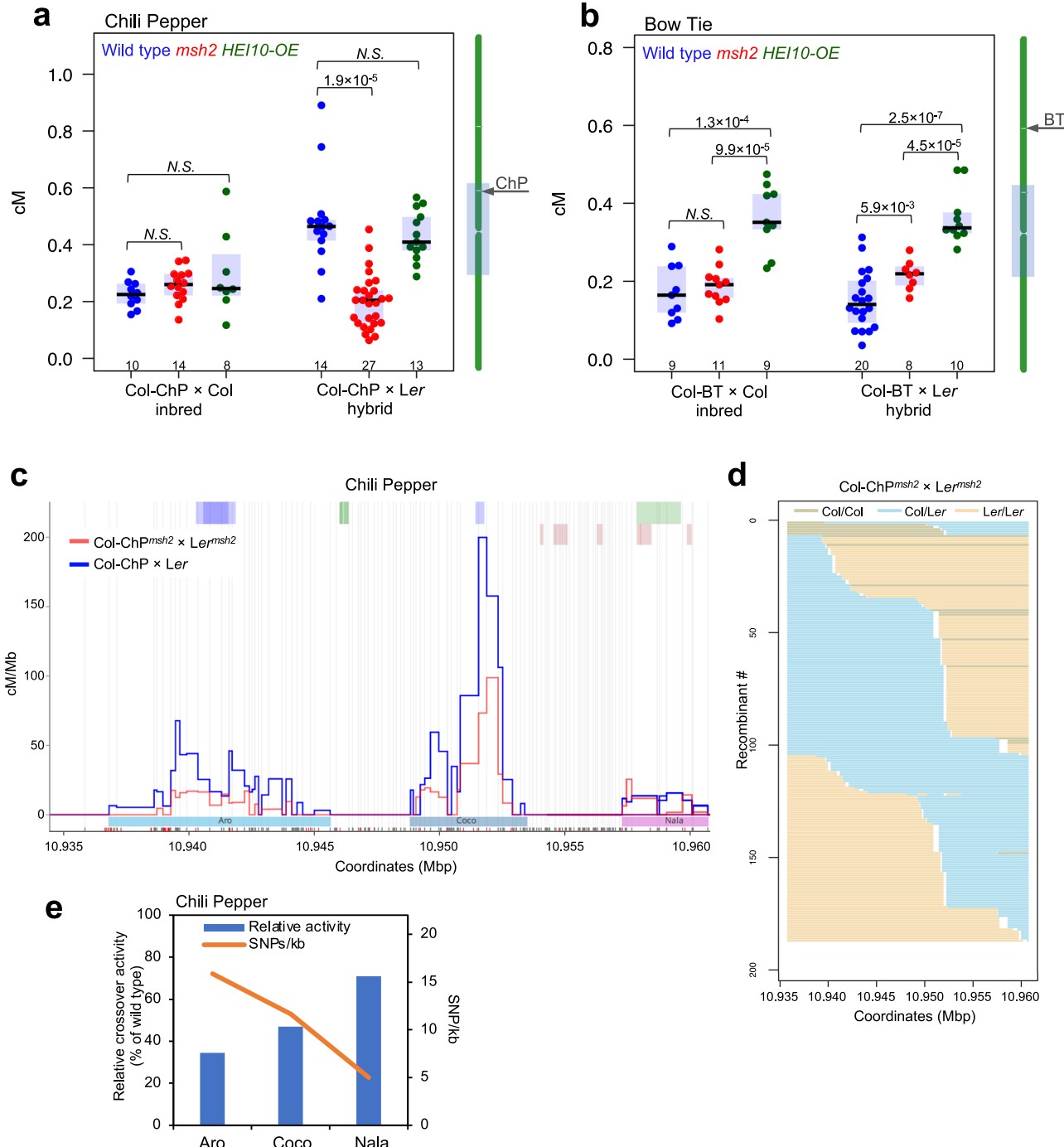

**Fig. 5 | Effect of SNP, the *msh2* mutation, and *HEI10* overexpression on CO frequency. a** CO frequency for Col-ChP × Col inbreds and Col-ChP × L*er* hybrids in the wild-type, *msh2*, and *HEI10* overexpression (*HEI10-OE*) genetic contexts. The center line of a boxplot indicates the mean; the upper and lower bounds indicate the 75th and 25th percentiles, respectively. Each dot represents a measurement from one individual. The numbers of individuals are also indicated below the boxplots. The two-sided *P* values were estimated by Welch's *t*-test. The chromosome 3 ideogram with the location of the ChP interval in relation to the pericentromere (gray box) is shown to the right. **b** As in **a**, but for the BT interval (Col-BT × Col inbreds and Col-BT × L*er* hybrids). **c** Comparison of the CO landscape within the ChP interval for the Col-ChP × L*er* wild type (blue) and Col-ChP*msh2* × L*ermsh2* (red).

COs are normalized to the measured ChP CO frequency. SNPs spaced at least 100 bp apart, shown as vertical gray lines, were used to determine the CO topology. The black and red x-axis ticks correspond to all Col/L*er* SNPs and InDels, respectively. **d** Genotypes of 188 Col-ChP*msh2* × L*ermsh2* recombinants. Each horizontal line corresponds to one recombinant with color-encoded genotype. The x-axis indicates coordinates within the ChP interval. **e** Relationship between SNP density and CO hotspot activity. Col-ChP × L*er* CO hotspot activity for the *msh2* mutant is represented as the proportion of the wild-type hotspot activity (blue bars). SNP density per each hotspot is shown as an orange line. Source data are provided as a Source Data file.

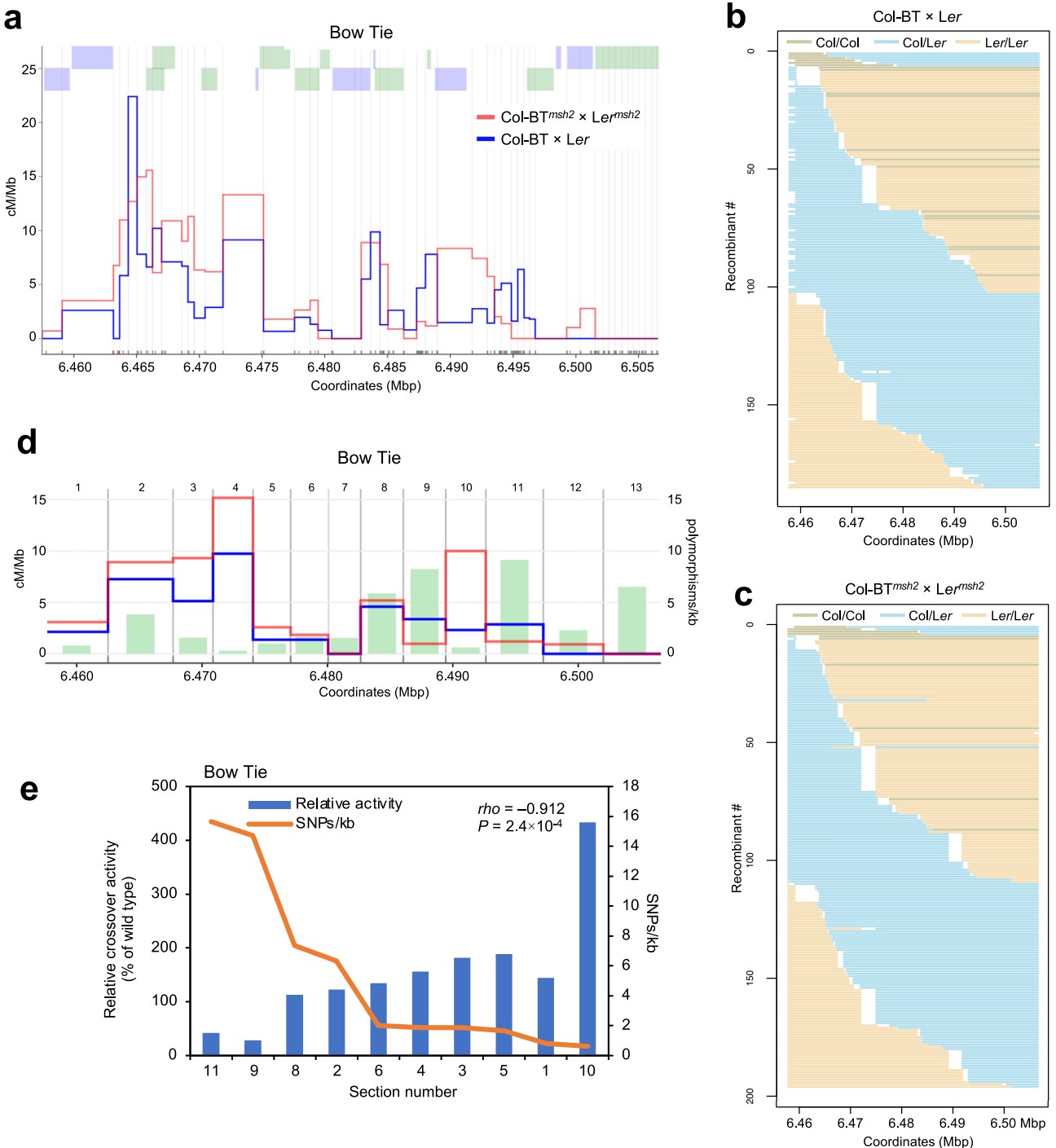

**Fig. 6 | Effects of the *msh2* mutation on CO distribution within the BT interval.**
**a** CO landscape within the BT interval for the Col-BT × L*er* wild type (blue) and Col-BT^*msh2* × L*er*^*msh2* (red). COs were normalized to the measured BT CO frequency. SNPs spaced at least 400 bp apart, shown as vertical gray lines, were used to determine the CO topology. The black and red x-axis ticks correspond to all Col/L*er* SNPs and InDels, respectively. Gene models are shown by green (forward) or purple (reverse) lines above the plot. **b** Genotypes of 185 Col-BT × L*er* recombinants. Each horizontal line corresponds to one recombinant with color-encoded genotype. The x-axes indicate coordinates within the BT interval. **c** As in **b**, but for 196 Col-BT^*msh2* × L*er*^*msh2* recombinants. **d** Col-BT × L*er* COs (blue; cM/Mb), Col-BT^*msh2* × L*er*^*msh2* COs (red; cM/Mb), and polymorphism density (green; SNPs + indels/kb) plotted in 13 sections of

the BT interval. COs were normalized to BT CO frequency measured for the wild type and *msh2*. The separation into 13 sections (numbers) based on polymorphism and CO pattern is indicated by vertical dashed lines. **e** Relationship between SNP density and CO activity. Col-BT × L*er* CO activity for *msh2* is represented as the proportion of the wild-type activity for the sections shown in **d** (blue bars). SNP density per section, is shown as an orange line. Sections #7, #12, and #13 were excluded from the analysis as they did not have COs in the wild type and/or *msh2*. The Spearman's rank correlation coefficient (*rho*) between SNP density and relative crossover activity with the two-sided *P* value is printed inset. Source data are provided as a Source Data file.

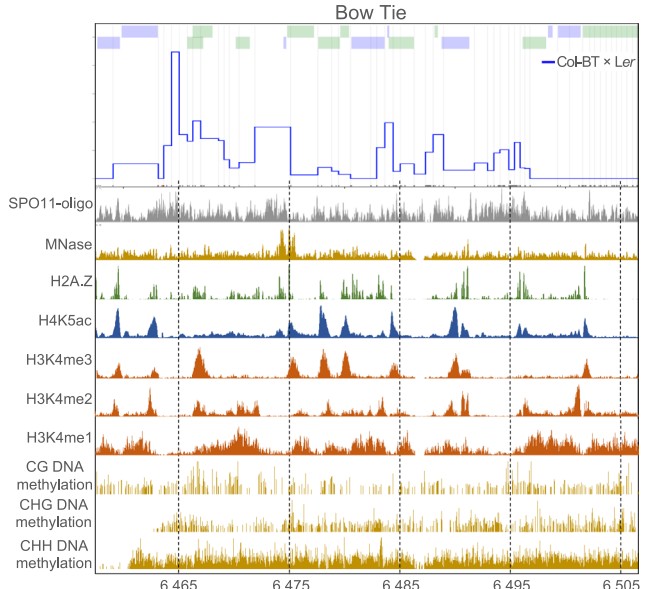

**Fig. 7 | Chromatin landscape in the BT interval.** Histograms showing normalized coverage values for SPO11−1-oligonucleotides[85], nucleosome occupancy (MNase-seq)[86], H2A.Z[86], H4K5ac[78], H3K4me1/me3[87], H3K4me2[88], and DNA methylation (BSseq) in CG, CHG, and CHH sequence contexts[89] were presented in relation to BT CO landscape for Col-BT × L*er*. Source data are provided as a Source Data file.

By crossing the R²-1 lines to L*er*, we obtained plants, where only the ChP interval were L*er*/L*er* homozygous while the remainder of the genome was Col/L*er* heterozygous (Fig. 8b). Interestingly, ChP CO frequencies were significantly lower in these plants than in ChP hybrids (Welch's *t*-test; $P = 4.5 \times 10^{-3}$; Fig. 8e). This effect disappeared in siblings carrying the *msh2* mutations (Fig. 8e). Hence, ChP shows a clear heterozygosity-homozygosity juxtaposition effect: When ChP is heterozygous in a fully homozygous genome, it shows an increase in CO frequency compared to hybrids and inbreds, while a decrease in CO frequency is observed when homozygous ChP is contrasted against the otherwise heterozygous genome (Fig. 8e and Supplementary Fig. 11).

In general, our study demonstrates that although crossovers tend to occur in hotspot centers that are devoid of polymorphism, SNPs in the immediate vicinity of hotspots are detected by MSH2, which stimulates the CO activity of these hotspots.

## Discussion

Here, we describe the seed-typing method for investigating the CO distribution with unprecedented resolution in two intervals located in the interstitial and pericentromeric regions of the chromosome. Particularly noteworthy is that we identified the pericentromeric interval, ChP, as one of the hottest regions in the entire genome based on the high-density CO map generated for the Col × L*er* cross (Fig. 1a)[26]. ChP is also the first pericentromeric region in Arabidopsis to be explored for high-resolution CO topology. ChP contains three gene-located hotspots that are clearly separated from each other (Fig. 1f). The central hotspot, Coco, had a recombination rate above 60 cM/kb and, to the best of our knowledge, is hotter than any other hotspot described in Arabidopsis to date[60–65]. A chromatin landscape analysis showed that Coco coincided with a region enriched for H3K4me2 but not H3K4me3 marks and that it was located within a long noncoding RNA gene. However, it remains unclear what makes this hotspot so active.

Structural rearrangements such as large insertions and deletions are often associated with local CO suppression[26,30–32]. However, this study dissects the influence of such big structural variations adjacent to meiotic hotspots on recombination. Surprisingly, we determined that a variation of <7 kb surrounding the hotspots did not significantly

affect their activity (Fig. 3). By contrast, we showed that deletions within a hotspot cause its partial inactivation (Fig. 3). This result was similar to the observation made for the mouse A3 hotspot, where the presence of three closely spaced small deletions define the CO refractory zone[32]. These results suggest that structural changes affect the frequency of recombination only when they are located within recombination hotspots. However, to what extent this is a universal phenomenon for all CO hotspots remains to be explored.

The ChP interval allowed us to investigate competition between hotspots, a phenomenon that has been reported in budding yeast, fission yeast, and mammals[66–70]. In Arabidopsis, silencing of the CO hotspot 3a via RNA-dependent DNA methylation (RdDM) results in a lack of compensatory effect in the neighboring hotspot 3b[63]. Similarly, the silencing of 3b does not affect the activity of 3a[63]. However, RdDM-mediated silencing induces chromatin changes that may have secondary consequences for hotspot activity and should therefore be interpreted with caution. Our results based on the inactivation of the hotspot by sgRNA-induced deletions support the lack of competition between immediate adjacent hotspots in Arabidopsis (Fig. 4).

Although the entire ChP interval is characterized by a high density of polymorphisms (18.7 InDels and SNPs per kilobase), COs mainly occurred in short sections ranging in length from several dozen to several hundred base pairs, which are almost identical between the two parents (Fig. 1f). This feature may be universal, as it was previously reported for other Arabidopsis hotspots and in other species such as mice[32,64,65]. Analysis of CO distribution in the Col-ChP × C24 cross with a slightly different polymorphism pattern revealed that the CO sites shifted to new, conserved (that is, lacking polymorphisms) chromosome sections, albeit within the same hotspots (Fig. 4a). Two important conclusions can be drawn from these observations: (1) The location of hotspots is independent of the presence of SNPs and (2) a chromosome section with a very limited number of interhomolog polymorphisms is required to efficiently form COs. The first of these conclusions is not surprising given previous reports indicating that the main determinants of CO sites in plants are open chromatin structure, active chromatin marks, and lack of DNA methylation[20,51,71,72]. However, the second conclusion is somehow unexpected given that Arabidopsis COs have been shown to have a local preference for heterozygous regions when juxtaposed to homozygous regions[39,50].

The fact that the detection of interhomolog polymorphisms during meiotic recombination depends on the mismatch detection system, the key element of which is the MSH2 protein[39], helped explain this phenomenon. Using the *msh2* mutant, we examined CO repair without polymorphisms (inbreds) and with polymorphisms (hybrids) for the centromere-proximal ChP interval and the interstitial BT interval. The effect of the *msh2* mutation on CO frequency was different between the two intervals (Fig. 5a, b), likely reflecting a CO redistribution along the chromosome in *msh2*, which is consistent with our previous findings[39]. Seed-typing revealed that hotspots in regions with a lower SNP density were used more frequently in *msh2* than in the wild type (Figs. 5c–e, 6). This observation indicates that MSH2 complexes actively stimulate CO repair in polymorphic hotspots.

To differentiate the polymorphism effect from potential chromosome-scale effects, we used a set of R² lines that carry heterozygous fragments only within ChP while the rest of the genome is homozygous. Surprisingly, we observed that the presence of polymorphisms in a single Coco hotspot (≤3.3 kb, 44 SNPs) was sufficient to stimulate COs over the entire ChP interval (26.3 kb; Fig. 8c). This result is similar to the effect described earlier in which the juxtaposition of heterozygous and homozygous regions is associated with increased CO frequency in heterozygous regions at the expense of neighboring homozygous regions (Supplementary Fig. 11)[39,50]. Significantly increased recombination was also reported in heterozygous regions of maize plants selfed for six generations[73]. Our results show that the heterozygosity/homozygosity juxtaposition effect is reflected at the

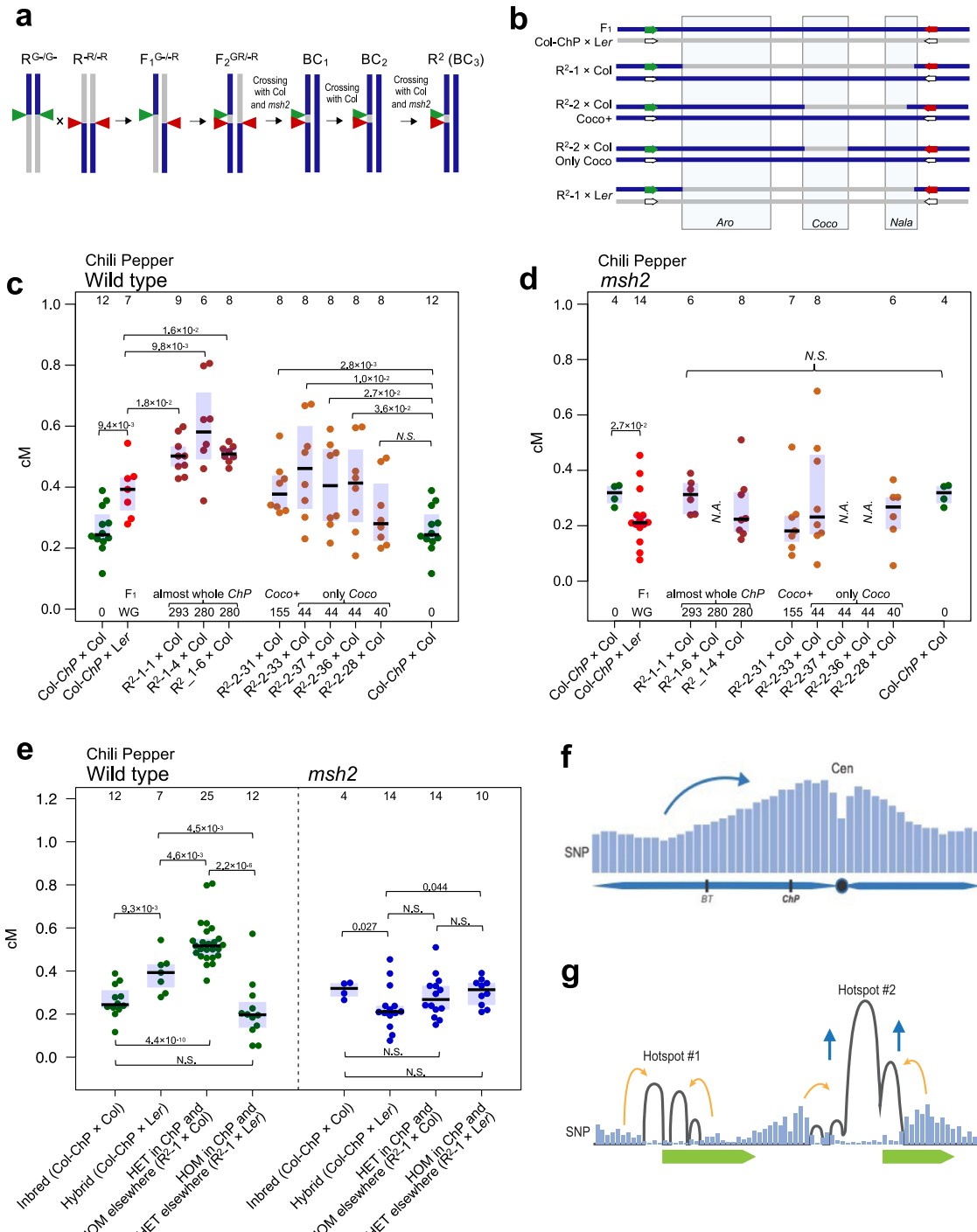

**Fig. 8 | Direct effect of polymorphisms on MSH2-dependent CO targeting within the ChP interval. a** Diagram illustrating the isolation of the R² lines. Double green (G-/G-) and double red (-R/-R) seeds were selected from the progeny of sequenced recombinants and crossed (F₁). Recombinant GR/-R F₂ seeds were visually selected and backcrossed to wild-type or *msh2* plants; at each generation, GR/-- seeds were selected. The BC₃ plants were used to measure ChP CO frequency. **b** Polymorphism pattern for different R² crosses. The chromosome carrying fluorescent reporters (green and red arrows) contains a segment inherited from the L*er* parent (gray). The Col-ChP × L*er* F₁ cross is shown for comparison. Hotspot positions are indicated in rectangles. **c** ChP CO frequency in R² lines. Each dataset corresponds to an independent R² line; The numbers below each bar indicate the number of SNPs differentiating homologous chromosomes. WG whole-genome. The center line of a boxplot indicates the mean; the upper and lower bounds indicate the 75th and 25th percentiles, respectively. Each dot represents one

individual. The two-sided *P* values were estimated by Welch's *t*-test. **d** As in **c**, but for the siblings of R²-1 lines in *msh2*. **e** ChP crossover frequency in Col/Col inbreds, Col/L*er* hybrids, lines Col/L*er* heterozygous (HET) in ChP while Col/Col homozygous (HOM) elsewhere, and lines L*er*/L*er* homozygous in ChP while Col/L*er* heterozygous elsewhere, in either wild type or *msh2*. A boxplot is defined as in **c**. Data for different R²-1 from **c** and **d** were presented together. The two-sided *P* values were estimated by Welch's *t*-test. **f, g** Model showing the effect of MSH2 on Arabidopsis crossovers. **f** At the chromosome-scale, COs are stimulated in polymorphic regions, close to the centromere (Cen). This crossover redistribution is specific for Class I COs and thus also depends on interference (blue arrow). Blue bars represent the local SNP density. **g** At the kilobase scale, hotspots located in SNP-rich regions are more active than neighboring hotspots at less polymorphic sites due to MSH2-dependent stimulation (blue arrows). However, COs occur mostly at polymorphism-free sites within each hotspot (orange arrows). Source data are provided as a Source Data file.

scale of a single hotspot. Importantly, these data offer a mechanical insight into this effect and provide an explanation for the apparent inconsistency with the observed preference of COs to occur in non-polymorphic intervals[65]. We observed that CO breakpoints invariably occur in the SNP-free sections of the hotspot in both the wild type and the *msh2* mutant. However, in the wild type, CO repair takes place more often in a hotspot whose immediate vicinity is SNP-rich. We propose that MSH2 heterodimers bind mismatches, forming hetero-duplexes that then recruit MLH1 complexes to induce a second cut, which triggers a CO repair (Fig. 8e–g). Other scenarios are also possible, including MSH2 heterodimers regulating branch migration[74] or stabilizing D-loops. Furthermore, interference causes significant CO redistribution at the chromosome-scale, as visualized by comparing wild-type and *msh2* CO patterns[39] (Fig. 8e).

As many as 20% of all plants are estimated to be predominantly selfing, including the model plant Arabidopsis and important monocot and eudicot crops like wheat, rice, and tomato[75,76]. Heterozygous regions in principally homozygous genomes that emerge from occasional outcrossing followed by many rounds of inbreeding are the only available sources of variation, both at the single individual and population scales[77]. Perhaps this is why meiotic recombination is targeted to heterozygous regions in selfing plants, as now evidenced in Arabidopsis. Our analysis shows how MSH2-dependent detection of inter-homolog polymorphism shapes the chromosomal CO pattern at the scale of single recombination hotspots. This feature may be of crucial importance in the evolution of self-pollinating plant species.

## Methods

### Plant material
Traffic lines used to obtain ESILs were created by ref. 57. Arabidopsis accessions (L*er*-0, An-1, Eri-1, Sha, Kyo, C24, Cvi-0, and Ct-1) and the *msh2* T-DNA insertion line (SALK_002708) was obtained from the Nottingham Arabidopsis Stock Centre (NASC).

### CRISPR-Cas9 mutagenesis
CRISPR-Cas9 technique was used to generate desirable genome modifications in ChP and in L*er*. It was based on designing a pair of sgRNAs to induce a more efficient mutagenesis[78,79]. Constructs carrying cassettes with sgRNAs cloned under *AtU3C* or *AtU6-26* promoters and Cas9 under *ICU2* promoter were introduced to plants via Agrobacterium-mediated transformation. Constructs used for the transformation of L*er* plants additionally contained a seed-expressed dsRed reporter marker for easier preselection of transformants[78,79]. Transformants were genotyped for the desired modifications and in the next generation construct-free plants with the heritable modifications were sequenced to confirm the length and the exact positions of the generated deletions.

In order to generate a deletion within the Coco hotspot in L*er*, a pair of sgRNAs (Coco_del1: 5′-gcctttgtgatttagtggcag-3′; Coco_del2: 5′-ccggtt aaaagtttaagac-3′) were designed to target cleavages at desired positions within the highest peak of recombination. To obtain a knock-out *msh2* mutant in L*er*, a pair of sgRNAs (msh2_1: 5′-agaacatgtaccgcagggat-3′; msh2_2: 5′-gcacctagagcaggagtcgca-3′) were designed to target the fourth exon of *MSH2*. The mutant was verified by RT-PCR amplification of the *MSH2* transcript from wild-type and mutant plants. Furthermore, the RT-PCR products from the *msh2-6* mutant was Sanger sequenced to confirm that the deletion produces a premature stop codon. CRISPR-Cas9 was also used to create pseudoreporter lines of ChP. To inactivate the fluorescence of the reporter cassettes, two pairs of sgRNAs were designed to target both dsRed (dsRed_1a: 5′- atcctgcaaactggaatcct-3′; dsRed_1b: 5′- aggattccagtttgcagga-3′ and eGFP (GFP_1a: 5′-aagttcagcgtgt ccggcg-3′; GFP_1b: 5′-ggcgagggcgatgccaccta-3′) coding sequences.

### Generation of extremely short interval lines
To obtain extremely short interval lines (ESILs), two lines carrying coding sequences of the seed-expressed fluorescent transgenes (eGFP

and dsRed) at the defined regions[57] were chosen and crossed with each other (Supplementary Table 1). The $F_1$ plants were selfed and the $F_2$ seeds were harvested. Seeds obtained from gametes in which a CO occurred between the reporters were preselected using epifluorescent stereomicroscope Lumar version 12 (Zeiss) equipped with dsRed and green fluorescent protein filters. As a result of CO, those seeds show the homozygous state of one reporter and a hemizygous state of the second one (GR/-R or GR/G-). The presence of the fluorescent cassettes in the plants grown from preselected seeds were confirmed by geno-typing using two primers flanking the reporter cassette mixed with the primer binding to the eGFP or dsRed coding sequences (Supplementary Table 7). If the lines are devoid of the particular cassette, the amplicons are obtained only with the left and right primers, and are between 300-600 bp. The $F_2$ plants were selfed and in the next generation, the homozygous ESILs were preselected under the epi-fluorescent stereomicroscope.

### Measurement of crossovers using a seed-based system
The recombination within the obtained ESILs was measured using a seed-based system, which was already described[50,80]. First, the homozygous ESILs were crossed with the non-color accessions and the pictures of $F_2$ seeds, derived from the entire plant, were acquired using an epifluorescent stereomicroscope (Zeiss). Each set of seed pictures included pictures in brightfield, and ultraviolet through filters for red and green fluorescence. The number of seeds was counted using CellProfiler program according to the protocol[80]. Due to low recombination rates within ESILs, the number of single-color recombinants was detected manually. To increase the accuracy of crossover measurements for such short intervals, all seeds from each plant were analyzed, which gives an average measurement of ~3000–5000 seeds per each biological replicate. The recombination frequency was presented in cM according to the formula: $RF = 100 \times ((N_G + N_R)/N_T)$, where $N_G$ is the number of only green seeds, $N_R$ is the number of only red seeds, and $N_T$ is the total number of seeds derived from the entire plant. Raw seed scoring data for all the measurements are presented in Supplementary Data 1.

### Preselection of recombinants based on the fluorescence
Recombinants, those seeds that have experienced the crossover between two reporter genes, were preselected manually based on the fluorescence intensity of the seeds. As a result of crossover, the recombinant seeds present single-color fluorescence (-R/-- or G-/--) or they are homozygous for one reporter and hemizygous for the second one (GR/-R or GR/G-).

### Seed-typing
DNA was extracted from the leaves of recombinant plants grown from preselected recombinant seeds. DNA extraction was performed as described in ref. 58. To provide only non-degraded, high-quality DNA for further steps of the seed-typing method, the extracted DNA was additionally purified using AMPure XP magnetic beads (Beckman-Coulter). The extracted DNA was mixed in a 1:1 ratio with magnetic beads and incubated for 5 min at room temperature. After the incubation, the samples were placed on the magnetic stand and incubated for 5 min. The supernatant was discarded and the beads were washed twice with 80% ethanol. DNA was eluted in the same volume of 10 mM Tris (pH 8.0) as the starting volume of the DNA. The quality of the DNA was checked in 1% agarose gel. Such prepared DNA was used as a template for long-range PCR (LR-PCR).

In LR-PCR, the whole ChP (26.3 kb) interval was amplified in three reactions (due to deletion at the 5′-end in L*er* and C24 when compared to Col, it was necessary to use different forward primers, therefore the interval in crosses with those accessions was amplified in four reactions) resulting in amplicons of 9–11 kb long. BT (49.1 kb) was amplified in five reactions with amplicons 10–11 kb long. The sequences of the primers

used for the amplifications are described in Supplementary Table 7. LR-PCR was performed using PrimeSTAR GXL Polymerase (TaKaRa Bio, Shiga, Japan). The PCR conditions were as followed: 0.2-10 ng of template DNA, 1.2 μL of 2.5 mmol/L primers, 3 μL buffer, 1.2 μL dNTP, 0.3 μL Polymerase, and distilled water to 15 μL, 30 cycles 98 °C 10 s, 68 °C 12 min. The efficiency of the amplification was verified in 1% agarose gel. Next, 10 μL of each PCR product amplified using the same DNA sample were pooled together. The DNA was purified (Clean-Up Concentrator, A&A Biotechnology) and eluted in 20 μL of 10 mM Tris (pH 8.0).

For library preparation, tagmentation was performed by mixing 1 μL of the pooled, purified PCR products with 1 μL of Tagmentation Buffer (40 mM Tris-HCl pH = 7.5, 40 mM MgCl$_2$), 0.5 μL of DMF (Sigma), 2.35 μL of Nuclease-free water (Thermo Fisher) and 0.15 μL of loaded, in-house produced Tn5 (Tn5 Transposase was prepared according to the protocol[81]). Loading Tn5 with the annealed linker oligonucleotides was followed[54]. The tagmentation reaction was incubated at 55 °C for 2 min and then stopped by adding 1 μL 0.1% SDS and incubation at 65 °C for 10 min. Amplification of the tagmented DNA was performed using the KAPA2G Robust PCR kit (Sigma) and custom P5 and P7 indexing primers. Each sample was amplified with the unique set of P5 and P7 primers as described in ref. 26. Subsequently, the libraries were visualized in a 2% agarose gel to group samples into pools of the comparable intensity of sheared DNA fragments between 450–700 bp. Each pool contained the same number of libraries. The concentrated libraries (Clean-Up Concentrator, A&A Biotechnology) were size selected in 2% agarose gel, performing electrophoresis for 2 h. Subsequently, DNA fragments of the desired size were excised and extracted from the gel using Gel Extraction Kit. The quality and quantity of the libraries were verified using TapeStation system (Agilent) and Qubit 2.0 fluorometer. Paired-end sequencing of 576 libraries per lane was performed on HiSeq X−10 instrument (Illumina).

To identify SNPs within the ChP or BT intervals between Col and other Arabidopsis accessions (Ler, C24), demultiplexed paired-end forward and reverse reads have been pooled and aligned to Col reference sequence with the use of bwa-mem algorithm from Burrows-Wheeler Aligner (BWA) software[82]. The Resulting BAM files have been sorted and indexed with the use of SAMtools v1.2[83]. SNPs were called using SAMtools and BCFtools[84]. Subsequently, SNPs with low coverage (<30 reads) and poor quality have been filtered out from the list. Individual sequencing libraries (243 samples from F$_2$ Ler × Col-ChP population, 177 samples from F$_2$ C24 × Col-ChP population, 209 samples from the F$_2$ Col-ChP × LerΔ#24 population, 187 samples from F$_2$ Ler$^{msh2}$ × Col-ChP$^{msh2}$ population, 160 samples from F$_2$ Ler × Col-BT population, 196 samples from F$_2$ Ler$^{msh2}$ × Col-BT$^{msh2}$ population) have been aligned to Col-0 ChP or BT sequences with the use of bwa-mem algorithm with default parameters and compared to previously generated SNP list with SAMtools and BCFtools. Characteristics of sequencing results for libraries in which COs were identified are shown in Supplementary Table 8. Custom R script enabled comparison of the percentage of reads associated with reference (Col) or variant nucleotides in Ler and C24 accessions. This has further allowed for genotype determination at particular SNP location and manual designation of CO breakpoint.

### Estimation of heterozygosity level in R$^2$ lines

To estimate the heterozygosity level in R$^2$ lines, genomic DNA from two randomly picked individuals representing each R$^2$ line was extracted as described (see Seed-typing) and used as a template to prepare libraries for sequencing (see Seed-typing). Paired-end sequencing was performed on the HiSeq X-10 instrument (Illumina). Demultiplexed raw data from paired-end sequencing have been aligned to the Col-0 reference. SNPs were called using SAMtools and BCFtools and compared to the list of SNPs from Col × Ler. Based on the results, sizes of Col/Ler heterozygous blocks for each chromosome were deduced. The *A. thaliana* genome size estimate of 125 Mb was used to calculate percent heterozygosity (Supplementary Table 6).

### Reporting summary

Further information on research design is available in the Nature Portfolio Reporting Summary linked to this article.

## Data availability

The seed-typing sequence data generated in this study have been deposited in the NCBI Sequence Read Archive (SRA) under the Bio-Project accession PRJNA882919. The Col-0 TAIR10 reference genome is downloaded from the TAIR database [https://www.arabidopsis.org/]. Genomic sequences of *Arabidopsis thaliana* accessions (An-1, C24, Cvi-0, Eri-1, Kyo, Ler, Sha) used in this study can be downloaded from https://1001genomes.org/projects/MPIPZJiao2020/index.html]. Col × Ler F$_2$ high-density crossover data were downloaded from FigShare [https://doi.org/10.25386/genetics.9733838]. Seed scoring raw data were provided in Supplementary Data 1. Source data are provided with this paper.

## Code availability

The related code is available at GitHub [https://github.com/LabGenBiol/ESILs].

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

## Acknowledgements

We thank Charles Underwood and Bertrand Llorente for discussions and comments on this manuscript and Beth Rowan for sharing a Col × Ler F$_2$ high-density crossover dataset. Natalia Glumińska is acknowledged for technical assistance. This work was supported by the Polish National Science Center (NCN) grants 2016/22/E/NZ2/00455 and 2020/39/I/NZ2/02464 to P.A.Z., 2021/41/N/NZ2/00340 to M.S.-L., the Foundation for Polish Science grant (POIR.04.04.00-00-5C0F/17-00) to P.A.Z., and the National Science Foundation grant (MCB-1614191) to R.S.P.

## Author contributions

M.S.-L. and P.A.Z. designed research; M.S.-L., W.D., J.D., and N.K. performed research; A.B. contributed new reagents; R.S.P. contributed new materials; W.D. performed the computational analyses; M.S.-L., W.D., J.D., N.K., and P.A.Z. analyzed data; and M.S.-L., W.D., and P.A.Z. wrote the paper.

## Competing interests

The authors declare no competing interests.
