## [Peer Review File · Nature Communications]

The effect of DNA polymorphisms and natural variation on crossover hotspot activity in *Arabidopsis* hybridsReviewers' Comments:

Reviewer #1:

Remarks to the Author:

The effect of polymorphisms on recombination is complex and not fully understood. Maja Szymanska-Lejman et al developed a new tool to study hot spots of COs in arabidopsis. They produced an impressive amount of data to assess the effect of DNA polymorphisms on these hot spots.

They showed that the insertion/deletion polymorphisms at the heart of the hotspot, but not adjacent ones, reduce hotspot activity, without compensation by adjacent hotspots. They further show that the absence of MSH2 leads to a decreased activity of polymorphic hotspots, suggesting that polymorphisms actively stimulate hotspot strength. These data and conclusions would be of great interest to the recombination field.

The data are solid and the interpretations justified, but I have some suggestions to improve the manuscript:

L37. dHJ cannot be repaired by SDSA. There is no support for this in ref 9

L40. dHJ were shown in cerevisiae to be mostly repaired as CO

L51 MUS81 is a nuclease, not a recombinase

L80. But COs are also promoted in these pericentromeric regions in absence of polymorphisms (Lian et al, ref 56 of this manuscript), suggesting that polymorphisms are not causal.

L134/Table S1. What is the genetic size of each interval?

L129 and the entire paragraph: Please provide statistical tests to support „significant“

L144. Most? Please provide a number.

L160. The supplementary figure 3 is very informative and should be included in the main figure

L167-182. The wording of this paragraph is confusing. The experiment tests for the effect of heterozygosity of the TDNA insertion (versus homozygosity for the TDNA, not absence), not for the presence of the T-DNA insertion. Please rephrase.

L194. Please provide a statistical test when comparing AN/Eri to other accessions

L213. Figure S6A should be in the main text. There may be compensation within the CoCo hotspot (increase at position 10.950). Please comment.

L243-244. Results on two intervals are insufficient to draw such a conclusion. Please rephrase.

L304. How was assessed the genotype of the rest of the genome?

L312 Add how hot is the coco hotspot. What is the absolute frequency of CO in this hotspot?

L360 In the absence of polymorphism, the proximal region is still very hot (Lian et al). How is this compatible with the conclusion that polymorphism is responsible?

Figure S10. It would be interesting to plot other features, as in figure S3.

Figure 6G. The represented model is highly speculative, and not supported by any data in the manuscript. There are many alternative models, as described in the text. Figure 6g should thus be deleted.

Reviewer #2:

Remarks to the Author:

This manuscript describes investigations of the effect of DNA sequence polymorphism on crossover (CO) patterning in highly recombinogenic region of Arabidopsis chromosome 3. The study itself is well conceived and executed. In particular, I am impressed how the authors developed a clever system to first enrich seed samples for recombinants before examining the CO patterns in details. The resulting data are certainly interesting. However, the extent to which data from a single interval can be extrapolated to the whole genomes is unclear. Is the ChP region representative for the whole genome? The authors mention that CO patterning in Arabidopsis is highly dependent on DNA methylation patterns but I cannot find in the manuscript information describing DNA methylation at this particular site. How the specific position of the ChP region on the chromosome affects recombination is also unclear. Also, ChP is one of the most CO active regions in the Arabidopsis genome – could this make its behavior special? I believe several claims in which the authors generalize their results should be

qualified:

Lines 193-197 – the phenomenon observed is interesting, but can we conclude that this is certainly a general rule?

Lines 311-313 – can it be concluded that DNA polymorphisms ALWAYS stimulate COs? Wouldn't large sequence differences have an opposite effect?

Lines 342-343: "our results... .. unequivocally confirm"

Also, the title and abstract should mention that the results refer to a single specific interval.

Other issues

In several instances, the authors state that differences were significant (for example on line 130) but the relevant test values are not listed.

Lines 347-350: I would be cautious with interpreting the apparently low correlation between DSBs and COs in plant studies. These studies used, as far as I can tell, different genetic backgrounds for DSB and CO assessments, which could have lowered the correlation levels.

Smaller issues:

Line 116: "ultra-high-density" – why not just "high density".

Line 123: the use of the term "inbreds" here is a bit confusing, as these lines are not exactly inbred.

Line 124: "polymorphism-dependent CO redistribution at the chromosome scale" – could this term be explained?

Line 320: I would avoid using the term "pericentromeric" as this can be confused with the pericentromeric regions of chromosome in crops, where CO rates are severely reduced.

Reviewer #3:

Remarks to the Author:

This study provides a detailed analysis of meiotic crossover (CO) distribution in a series of hotspots in *Arabidopsis thaliana*. This unique dataset allows them to provide an exhaustive and original analysis of the relationships between SNPs density and the distribution of meiotic crossovers in polymorphic hotspots. The main conclusions are that crossovers (COs) occur preferentially within polymorphic hotspots and that this effect depends on the mismatch repair gene MSH2. This is a new insight of this study, and it is well documented and analyzed.

The authors developed a seed-typing method that enables massive parallel fine-mapping of crossovers by sequencing. They performed an incredible amount of work in selecting a series of new regions containing clusters of hotspots. The Chp region is of high interest because this is the first time that a cluster of hotspots has been localized and analyzed in the pericentromeric region of a plant.

Pericentromeres are highly polymorphic between *Arabidopsis* accessions and are particularly suitable for analyzing the effect of SNPs on CO hotspot activity. They performed crosses between the Col accession and a series of accessions with diverse polymorphisms in the Chp region. All hybrids, except one, had a significantly higher CO frequency in the Chp region than in the corresponding inbred region. This result, performed at the hotspot scale, confirmed a series of genome-wide analyses. Moreover, the loss of MSH2 results in a decrease in COs formation in the most polymorphic hotspots. This is a very intriguing result, as inactivation of MSH2 in the budding yeast *S. cerevisiae* gives the opposite result. To consolidate their results, the authors constructed new lines in which only the Chp region was heterozygous, to eliminate the effect of genome-wide heterozygosity. Using this unique

material, the authors confirmed that local polymorphisms stimulate CO formation. This detailed analysis provides new insights into the role of mismatch repair genes during meiotic recombination. It is of primary interest not only for people working on meiosis, but also for people working on the role of mismatch repair during recombination and DNA damage repair.

Minor points

- I do not find the way CO rates are calculated. Why do the authors not use a simple metric in cM/Mb ? I would also like to know the number of CO they have per interval in the sup data
- - in the same range of idea line 157 they mention that Coco is at 7.77 cM/Mb but lane 323 it is 60 cM/kb
- For the hotspot's competition, I suggest that the authors could moderate their conclusions given in the abstract and the discussion. The fact that they do not see competition in their dataset does not signify that other hotspot outside the interval studied could not be impacted . In *S. cerevisiae* the effect is seen in one third of a chromosomal arm.
- - same comment for the sentence lines 334-335 and for the sentence in the abstract, they do not see an effect of an insertion located just outside the cluster of hotspots but it is difficult to draw a general conclusions from one observation.
- They obtain a new allele *msh2* by performing CrispR Cas9 in the Ler background. How do they validate the allele ? have they looked at meiotic spreads ? pollen viability? Transcription ?

Reviewer #4:

Remarks to the Author:

DNA polymorphism effect on hotspot activity

The manuscript from the Ziolkowski lab uses a powerful seed-typing method to identify and fine-map crossovers in two crossover hotspot intervals. Using this method they undertake a considerable amount of work investigating the effect of polymorphism on CO rates in these intervals. Within the ChP and BT intervals CO rates were highest in regions containing fewer SNPs, but total COs were increased in the ChP interval in hybrids compared to pure lines. In contrast there was no change in COs in the BT interval in hybrids compared to pure lines. These results suggest an intriguing possibility that SNPs differentially affect CO designation at different scales – with SNPs stimulating COs at the (10s of) kilobase scale, but with COs being placed in small regions of homozygosity within these broader regions of high SNP density. Unfortunately, the experiments do not quite go far enough to convince me of this possibility.

The authors also showed that deletions within hotspots decreased their activity, but no compensatory increase was observed in neighbouring hotspots. In addition, large structural polymorphisms adjacent to the hotspots did not affect CO rates.

There is a lot to commend in this paper and it represents a considerable amount of work. However, I do have several issues. The first major issue is that in figure 1f and all similar representations, CO rates are not normalised by window length. This can be very misleading, for example if CO rates were uniform across the whole interval, regions of lower SNP density would artificially appear to have more COs because the COs occurring in a larger window (i.e. distance between adjacent SNPs) are being summed. The y-axis value would be proportional to the interval between adjacent SNPs. To get around this issue the y-axis value in 1f (and all similar plots) should be given in cM/Mb – this may require splitting the interval up into windows of similar size (as in 5d).

The representation in 5d is also problematic, as all COs are being recorded at the mid-point of the two flanking SNPs which gives an artificially high sense of the CO resolution. Again, for Figure 5d the y-axis units should be cM/Mb, with a single value per window given.

My second major issue is one of over interpretation. "General" rules are stated, while the authors have only looked at two intervals and these two intervals show different results. The authors propose a plausible explanation for this, based on redistribution of COs due to differences in relative SNP density, however they have not performed the required experiments to convince me of this explanation.

Undertaking CO analysis in R2 BT lines and in R2 BTmsh2 and R2 ChPmsh2 lines would (if the results are in line with the hypothesis) make me much more convinced of the proposed explanation.

While I have some issues with the manuscript, the question being addressed is an important one for the community and the field is full of apparently contradictory findings both on the influence of SNPs on recombination and the role of MSH2. There is much in this manuscript that helps shed light on these factors and how they interact to shape CO distributions.

Additional specific points are given below.

Line 42: is repaired by -> are repaired as

Line 73: I think there is still much to be learned regarding the relationship between SNPs and COs – and some of this could be brought out more in the intro. The lack of a juxtaposition effect in msh2 is supportive of SNPs having an MSH2 mediated pro-crossover effect. On the other hand, it has recently been reported by the Mercier group (Lian et al 2022, Nat. Comm.) that the megabase-scale crossover landscape is largely independent of sequence divergence with recombination distributions in Arabidopsis pure lines being very similar to those of Intra-specific hybrids (e.g. highest recombination rates in pericentromeric regions in pure lines). This supports the notion that much of the association of CO rates and SNPs is due to mutagenic effects of recombination rather than a pro-crossover influence of SNPs.

Line 144: This seems excessively high? Or is this combined depth for all recombinants rather than depth per amplicon?

Line 148 – Figure 1f: Y-axis value should be normalised by window length e.g. cM/Mb, otherwise I cannot tell if regions with a high y-axis value truly have higher recombination rates, or whether it's just a case of more recombination events being detected because the regions are larger (i.e. greater distance between adjacent SNPs).

In order to really see the recombination rate across the region in a comparable way, the y-axis needs to be converted to cM/Mb. For very small intervals between adjacent SNPs this will likely generate a very noisy plot, so it may make more sense to plot the cM/Mb value across the region using a set window length, rather than intervals of variable length determined by the spacing of adjacent SNPs.

Line 179: Perhaps I am mis-understanding something here, but I don't follow the logic. Why would the presence of the fluorescent proteins affect recombination?

For dsRED the T-DNA is still present so presumably the transgene is still transcribed even if it doesn't result in a functional mRNA? Given transcriptionally active genomic regions have higher recombination rates mightn't the (non-functional) T-DNA still affect COs? How does the recombination rate in a Col-ChP x Ler measured using the fluorescent reporters correspond to the CO rate determined in other published genome wide recombination analyses? E.g. Rowan et al. Wouldn't this also answer the question?

Or is this more about whether plants are hemizygous for the T-DNA insertion – the idea being that there is structural polymorphism between the two homologs in Col-ChP X Col, but not in Col-ChP X Col-ΨChP? If the GFP cassette is completely deleted in Col-ΨChP doesn't this mean that there is still structural polymorphism in both Col-ChP X Col and Col-ChP X Col-ΨChP? In which case, my

interpretation would be that it is not possible to determine whether the T-DNA influences CO rates. Am I missing something?

The nature of the CRISPR mutations is also very unclear. First in the text it states the authors generated a "pseudo-reporter line in which the reporter cassettes were preserved but produced nonfunctional fluorescent proteins". Later in the text it says the GFP cassette was "largely removed" and in the Figure S4 legend it states the GFP cassette was completely removed. In the legend to Figure 2 it says GFP has a missense mutation. Which is it?

For dsRED the main text and Figure S4 state that there is a frameshift mutation, while in the Figure 2 legend it says dsRED has a missense mutation. Again, which is it?

Figure S4 needs more detail, what sequences are being aligned here? Are the chromatograms being aligned with wild-type genomic DNA for GFP and the T-DNA sequence for RFP? This needs better labelling.

Line 192: Was this tested statistically? Needs to be included in text, figure or table if claims of significance made. Perhaps a supplementary table with pair-wise significance tests?

Line 240: "Similar to ChP, BT inbreds showed no change from the wild type" – Presumably the BT inbreds referred to here are msh2 mutants? Would be good to make this explicit.

Line 255: This general statement is not supported by the data. It is only true for the ChP hotspot, not for the BT hotspot (Figure 4a-b). This should be modified to refer only to the ChP hotspot. Or is this referring to finer scale hotspots within the ChP and BT intervals??

If this is the case, I'm not sure how it is possible to show that SNPs stimulate COs, given that it is impossible to work out the distribution of CO events within an interval in the absence of any polymorphism.

It is possible to make the argument that MSH2 promotes COs in a SNP density dependent manner at the individual hot-spot scale (Fig 4e and Figure 5e). It does not follow however that SNPs promote COs at the hotspot scale, there is no apparent correlation between SNP density and CO rate in figures 4C and 5a/5d, though this has not been looked at explicitly. To do this it would be best to split the intervals up into windows of similar size (as has been done for BT in figure 5d) and then do a scatter plot comparing SNP density with CO rate measured in cM/Mb. It would then be possible to test if there is any correlation between SNP density and CO rate i.e. to see if "SNPs stimulate COs at the hotspot scale".

I also do not like the representation in figure 5d which gives an artificially high sense of the resolution of CO mapping. The narrow widths of the blue and red peaks imply a very narrow distribution of CO positions where in reality the resolution of CO position is limited by SNP density. For example, the large peak in interval 4 is high because all COs in a 5kb region between adjacent SNPs are being recorded at the midpoint between the two SNPs. Also what does the metric CO percent correspond to? Is this the proportion of all BT COs that occur in this window? This is problematic given that the windows are of different sizes. Normalising by window length (e.g. cM/Mb). Also, a single CO value (in cM/Mb) and SNP density value (polymorphisms/kb) should be used for each window.

Line 291: This ambiguous, it sounds like you are looking at recombination events that occurred in the 4.5 Mb genomic region in-between BT and ChP.

Line 311++: Again, I do not agree with this statement. For one interval (ChP) there was higher recombination in a hybrid compared to pure lines, for a second interval (BT) there was no change in recombination in a hybrid compared to pure lines. It is impossible to make a "general" statement

when two intervals were analysed and they both responded differently to polymorphism and *msh2* mutation (Figure 4a / 4b).

One possibility is that in a hybrid the effect of higher SNP density in pericentromeric regions (e.g. ChP) dominates any potential stimulatory effect of SNPs in chromosome arms. Indeed, a wealth of recent papers show that ZMM crossover numbers are set by HEI10 dosage, so SNP density could at most be expected to cause a redistribution of COs toward regions of higher SNP density rather than stimulate additional COs. In fact, as Ler has a weaker HEI10 allele than Col, hybrids have less COs than pure Col (Lian et al 2022, Nat. Comm.).

Line 365: I do not find this surprising given the same regions that show highest levels of polymorphism between accessions also show the highest levels of recombination in pure lines (Lian et al 2022, Nat. Comm.). This suggests there is some other factor (i.e. not inter-homolog polymorphisms) is contributing to chromosomal scale CO distributions in Arabidopsis.

Line 374: Check wording here.

Line 377: Sorry I don't follow this reasoning. Was there any statistical test done comparing CO rates in *msh2* pure lines and hybrids? These don't appear on the Figure referenced. In the absence of *msh2* there appears to be no change in COs between pure lines and hybrids for both ChP and BT? What is the relevance of the HEI10-OE lines?

To be fully convinced that SNPs generally stimulate COs (and that it is not largely a position effect, or an effect restricted to the peri-centromeric regions) I would need to see data for R2 lines for the BT interval. If SNPs do stimulate COs then it would be expected that in contrast to a Col/Ler hybrid (where SNP density in BT is lower than pericentromeric regions), CO rates would increase in a BT R2 line (SNP density in the BT region being higher than in pericentromeric regions).

To be convinced that the effect is MSH2 dependent (and not resulting from a SNP independent CO redistribution in *msh2* mutants) I would also need to see results for a BT and ChP R2 line in an *msh2* background. If the authors' hypothesis is correct, then both ChP and BT R2 *msh2* lines should show no change in CO compared to inbred wt.

Dear reviewers,

Our responses to your comments are highlighted in blue type.

All changes in the revised Manuscript File and the SI Appendix have been highlighted in blue.

Reviewer #1 (Remarks to the Author):

The effect of polymorphisms on recombination is complex and not fully understood. Maja Szymanska-Lejman et al developed a new tool to study hot spots of COs in arabidopsis. They produced an impressive amount of data to assess the effect of DNA polymorphisms on these hot spots.

They showed that the insertion/deletion polymorphisms at the heart of the hotspot, but not adjacent ones, reduce hotspot activity, without compensation by adjacent hotspots. They further show that the absence of MSH2 leads to a decreased activity of polymorphic hotspots, suggesting that polymorphisms actively stimulate hotspot strength. These data and conclusions would be of great interest to the recombination field.

We thank the Reviewer for appreciating our work.

The data are solid and the interpretations justified, but I have some suggestions to improve the manuscript:

L37. dHJ cannot be repaired by SDSA. There is no support for this in ref 9

L40. dHJ were shown in cerevisiae to be mostly repaired as CO

L51 MUS81 is a nuclease, not a recombinase

We thank the reviewer for these comments, the manuscript has been accordingly corrected.

L80. But COs are also promoted in these pericentromeric regions in absence of polymorphisms (Lian et al, ref 56 of this manuscript), suggesting that polymorphisms are not causal.

The reviewer is right, and we chose to delete this sentence and modify the whole paragraph.

L134/Table S1. What is the genetic size of each interval?

We now included the genetic size for each interval in Supplementary Table 1.

L129 and the entire paragraph: Please provide statistical tests to support „significant“

In the case to which the reviewer is referring, it is difficult to perform a proper statistical analysis as we are dealing with two different types of data: single value for the mean of genomic recombination calculated from the genetic maps versus the measurement using reporter lines. Therefore, we have replaced the word "significantly" with "substantially". In all other cases where we compare values, we have performed appropriate statistical tests and reported the results in figures. In the new version of the manuscript, some of these statistical test results are additionally emphasized in the text of the manuscript.

L144. Most? Please provide a number.

We included the numbers as requested.

L160. The supplementary figure 3 is very informative and should be included in the main figure

Thank you for this suggestion. We incorporated the revised version of the *ChP* browser view into the manuscript as Fig. 2.

L167-182. The wording of this paragraph is confusing. The experiment tests for the effect of heterozygosity of the TDNA insertion (versus homozygosity for the TDNA, not absence), not for the presence of the T-DNA insertion. Please rephrase.

We agree with the reviewer that this part was not clearly written. We revised the text to make it easier to follow.

L194. Please provide a statistical test when comparing AN/Eri to other accessions

The results of a statistical test were indicated on Fig. 2d. We added a reference for this in the text to facilitate finding this result.

L213. Figure S6A should be in the main text. There may be compensation within the CoCo hotspot (increase at position 10.950). Please comment.

Although the comparison of *seed-typing* data from Col × C24 and the Col × Ler suggests compensation, in our opinion it is a technical consequence of a different pattern of polymorphisms in those line. there is a long subinterval devoid of polymorphisms around position 10.950 Mb in Col × C24 when compared to Col × Ler . In the previous presentation of the results, where the CO number was not normalized to the length of the subinterval, it gave the impression of a local increase in CO frequency. In line with the comments of the other reviewers, in the revised manuscript we presented data normalized to the length of subintervals (in cM/kb) and in this approach, the apparent compensation effect disappears.

L243-244. Results on two intervals are insufficient to draw such a conclusion. Please rephrase.

We have toned down this conclusion to read: “These results suggest that MSH2-dependent changes in recombination levels **observed for ChP and BT in hybrids may be affected** by polymorphism-dependent CO redistribution at the chromosome scale.”

L304. How was assessed the genotype of the rest of the genome?

To verify how efficient was backcrossing of R² lines, we sequenced two individuals representing each R² line using Illumina HiSeq X-10 platform with a coverage of around 2×. Demultiplexed raw data from paired-end sequencing have been aligned to Col-0 reference and compared to the Col×Ler SNP list. Analysis of the sequencing results showed that only the minor fragments of the chromosomes are Col×Ler heterozygous. We included this result in the revised manuscript as a Supplementary Table 6. The description of the procedure has been included in Methods.

L312 Add how hot is the coco hotspot. What is the absolute frequency of CO in this hotspot?

The absolute CO frequency in the Coco hotspot is 0.288 cM. Now we included this information in Supplementary Table 3.

L360 In the absence of polymorphism, the proximal region is still very hot (Lian et al). How is this compatible with the conclusion that polymorphism is responsible?

The reviewer is right that high crossover frequency in proximal chromosome regions is independent on interhomolog polymorphism. We do not deny this observation anywhere in our work. Moreover, in the quoted excerpt we conclude that the location of recombination hotspots is independent of polymorphism and that the crossover can only be formed in sections devoid of polymorphisms. Our observations only show that the presence of SNPs in the immediate vicinity of hotspots stimulates their activity, but the location of the hotspots remains unchanged. The work of Lian et al. 2022 does not allow for such resolution, therefore they could not conclude about the influence of SNPs on the relative activity of individual hotspots. In conclusion, both works are complementary and not contain contradictory conclusions.

Figure S10. It would be interesting to plot other features, as in figure S3.

Thank you for this suggestion. We incorporated the revised version of the *BT* browser view into the manuscript as Fig. 7.

Figure 6G. The represented model is highly speculative, and not supported by any data in the manuscript. There are many alternative models, as described in the text. Figure 6g should thus be deleted.

We removed the model presented on Fig. 6g as suggested.

Reviewer #2 (Remarks to the Author):

This manuscript describes investigations of the effect of DNA sequence polymorphism on crossover (CO) patterning in highly recombinogenic region of Arabidopsis chromosome 3. The study itself is well conceived and executed. In particular, I am impressed how the authors developed a clever system to first enrich seed samples for recombinants before examining the CO patterns in details. The resulting data are certainly interesting. However, the extent to which data from a single interval can be extrapolated to the whole genomes is unclear. Is the ChP region representative for the whole genome? The authors mention that CO patterning in Arabidopsis is

highly dependent on DNA methylation patterns but I cannot find in the manuscript information describing DNA methylation at this particular site.

We agree that some of our data is limited to the *ChP* region and we made appropriate corrections in the text where needed. However, we would like to point out that the most important result presented in this paper, which is the effect of SNPs on CO occurrence, was examined in two intervals, i.e. *ChP* and *BT*. Despite a completely different chromosomal location (*ChP*, pericentromeric, *BT*, located in the middle of the chromosomal arm), both intervals show the same effects when the crossover topology is compared between wild type and the *msh2* mutant: polymorphism stimulates recombination activity at the hotspot scale. Moreover, the new data for the *ChP* interval (Fig. 8e and Supplementary Fig. 11) added by us in the revision show that the observed effect is similar in essence to the heterozygosity-homozygosity juxtaposition effect, which has recently been described for the subtelomeric interval (Blackwell et al., 2020, doi: 10.15252/embj.2020104858). Therefore, we believe that the relationships we have described between SNP polymorphism and meiotic recombination are universal.

In the revised version of our work, we also added data on the DNA methylation profile within the studied regions (Fig. 2 & 7).

How the specific position of the ChP region on the chromosome affects recombination is also unclear. Also, ChP is one of the most CO active regions in the Arabidopsis genome – could this make its behavior special? I believe several claims in which the authors generalize their results should be qualified:

We agree that data from two hotspots for the *msh2* mutant and from one hotspot for the effect of structural variation on crossover formation cannot be extrapolated to the entire genome. We therefore thoroughly rewritten the manuscript to underline the fact that similar analyzes of additional hotspots are required to come to more general conclusions.

Lines 193-197 – the phenomenon observed is interesting, but can we conclude that this is certainly a general rule?

This conclusion was rephrased to make it clear that it refers to the *ChP* interval.

Lines 311-313 – can it be concluded that DNA polymorphisms ALWAYS stimulate COs? Wouldn't large sequence differences have an opposite effect?

Indeed, in earlier works, especially in Blackwell et al., 2020, doi: 10.15252/embj.2020104858, we observed a parabolic relationship between crossover frequency and SNP density. Initially, these two features show a positive correlation, but higher polymorphism density associates with reduced crossover frequency. However, the data used to perform the analysis of Blackwell et al. comes from hundreds of crossover midpoints, each containing a different SNP distribution. Consequently, information about the crossover topology is lost.

In the case of the results included in the present study, the data comes from only several hotspots in *ChP* and *BT* intervals, but for each of them we have precise information on the distribution and frequency of CO in the background of wild-type and the *msh2* mutant. Thanks to this, we could investigate the topology of hotspots and we could conclude about the influence of SNP on crossover formation. For *ChP*, we have also provided the information for the R^2 lines, where effects of polymorphisms located outside the interval are eliminated (new Supplementary Table 6).

Based on our *seed-typing* results, we were also able to observe that the crossover breakpoint site is ALWAYS practically devoid of polymorphisms. This could be also observed in previous observations made using the pollen-typing method developed by Christine Mezard's group and used by others (e.g., Serra et al. 2018, doi: 10.1371/journal.pgen.1007843; Choi et al. 2016, doi:10.1371/journal.pgen.1006179). However, our analyzes show that an important factor for hotspot activity is the SNP density in the immediate vicinity of hotspots. This result could not be captured using the approach used in Blackwell et al. 2020, where a specific averaging of data has been used. Therefore, our data explains the paradox that on the chromosome segment scale polymorphism stimulates CO (heterozygosity-homozygosity juxtaposition effect; Ziolkowski et al. 2015, doi:10.7554/eLife.03708), while on the CO hotspot scale CO always occurs in regions devoid of polymorphisms.

Saying so, we agree that this is not clear in the manuscript. Therefore, we have emphasized this observation more fully in the discussion.

Lines 342-343: "our results... .. unequivocally confirm"

As we referred to the previous studies on hotspot 3a and 3b, we have now changed this to read: “Our results based on inactivation of the hotspot by sgRNA-induced deletions **supports** the lack of competition between adjacent hotspots in *Arabidopsis*.”

Also, the title and abstract should mention that the results refer to a single specific interval.

We made appropriate changes to the abstract, but it seems to us that the title is very general and there is no need to change it. Also, we have studied two chromosomal regions, *ChP* and *BT*, which have different chromosomal location.

Other issues

In several instances, the authors state that differences were significant (for example on line 130) but the relevant test values are not listed.

We thank the reviewer for this comment. In the case to which the reviewer is referring, it is difficult to perform a proper statistical analysis as we are dealing with two different types of data: single value for the mean of genomic recombination calculated from the genetic maps versus the measurement using reporter lines. Therefore, we have replaced the word "significantly" with "substantially". In all other cases where we compare values, we have performed appropriate statistical tests and reported the results in figures. In the new version of the manuscript, some of these statistical test results are additionally presented in the text of the manuscript.

Lines 347-350: I would be cautious with interpreting the apparently low correlation between DSBs and COs in plant studies. These studies used, as far as I can tell, different genetic backgrounds for DSB and CO assessments, which could have lowered the correlation levels.

As suggested by the reviewer, we have removed an excerpt from the discussion regarding the potential reasons for the lack of competition between *Arabidopsis* hotspots.

Smaller issues:

Line 116: “ultra-high-density” – why not just “high density”.

This is the term used by the authors of the publication we are referring to, also used in the title of the publication, so we decided to keep it.

Line 123: the use of the term “inbreds” here is a bit confusing, as these lines are not exactly inbred.

The reviewer is right on this point. Therefore, in the mentioned place of the manuscript we explained the difference between ESIL crosses with Col and crosses with other accessions. However, for the sake of simplicity, and also due to the fact that such nomenclature is used in publications of other groups when referring to fluorescently tagged lines (e.g., Girard et al., 2015, doi:10.1371/journal.pgen.1005369; Nageswaran et al., 2021 doi: 10.1038/s41477-021-00889-y; Kim et al., 2022, doi: 10.15252/embj.2021109958), we have chosen to keep the term "inbreds".

Line 244: “polymorphism-dependent CO redistribution at the chromosome scale” – could this term be explained?

We apologize for this very vague term. In the revised manuscript we remove it and used more careful wording instead.

Line 320: I would avoid using the term “pericentromeric” as this can be confused with the pericentromeric regions of chromosome in crops, where CO rates are severely reduced.

We have changed the text to emphasize that this is the first pericentromeric region in *Arabidopsis* studied with this resolution.

Reviewer #3 (Remarks to the Author):

This study provides a detailed analysis of meiotic crossover (CO) distribution in a series of hotspots in *Arabidopsis thaliana*. This unique dataset allows them to provide an exhaustive and original analysis of the relationships between SNPs density and the distribution of meiotic crossovers in polymorphic hotspots. The main conclusions are that crossovers (COs) occur preferentially within polymorphic hotspots and that this effect depends on the mismatch repair gene MSH2. This is a new insight of this study, and it is well documented and analyzed.

The authors developed a seed-typing method that enables massive parallel fine-mapping of crossovers by sequencing. They performed an incredible amount of work in selecting a series of new regions containing clusters of hotspots. The Chp region is of high interest because this is the first time that a cluster of hotspots has been localized and analyzed in the pericentromeric region of a plant. Pericentromeres are highly polymorphic between Arabidopsis accessions and are particularly suitable for analyzing the effect of SNPs on CO hotspot activity. They performed crosses between the Col accession and a series of accessions with diverse polymorphisms in the Chp region. All hybrids, except one, had a significantly higher CO frequency in the Chp region than in the corresponding inbred region. This result, performed at the hotspot scale, confirmed a series of genome-wide analyses. Moreover, the loss of MSH2 results in a decrease in COs formation in the most polymorphic hotspots. This is a very intriguing result, as inactivation of MSH2 in the budding yeast *S. cerevisiae* gives the opposite result. To consolidate their results, the authors constructed new lines in which only the Chp region was heterozygous, to eliminate the effect of genome-wide heterozygosity. Using this unique material, the authors confirmed that local polymorphisms stimulate CO formation. This detailed analysis provides new insights into the role of mismatch repair genes during meiotic recombination. It is of primary interest not only for people working on meiosis, but also for people working on the role of mismatch repair during recombination and DNA damage repair.

We are pleased that the reviewer appreciates the research presented in the study.

Minor points

- I do not find the way CO rates are calculated. Why do the authors not use a simple metric in cM/Mb ? I would also like to know the number of CO they have per interval in the sup data

We apologize for presenting CO rates in the plots not clearly enough. The problem with using cM/kb was that some short sections between adjacent SNPs resulted in very noisy plots. In the revised manuscript, we solved this problem by combining SNPs separated by less than 100 bp (for the *ChP* interval) or less than 400 bp (for the *BT* interval). This allowed us to present the data in a conventional way (cM/kb) with relatively little loss of resolution. The information on the number of mapped COs per interval were indicated in the Supplementary Table 9.

- - in the same range of idea line 157 they mention that *Coco* is at 7.77 cM/Mb but lane 323 it is 60 cM/kb

We thank for detecting this inconsistency, which has been corrected in the revised manuscript. We double checked our calculations and the correct cM/kb value for the *Coco* hotspot currently given in the text and in Supplementary Table 3 is 62.14 cM/kb.

- For the hotspot's competition, I suggest that the authors could moderate their conclusions given in the abstract and the discussion. The fact that they do not see competition in their dataset does not signify that other hotspot outside the interval studied could not be impacted . In *S. cerevisiae* the effect is seen in one third of a chromosomal arm.

In the abstract, introduction, results and discussion, we have softened the conclusions on hotspot competition by emphasizing the fact that they only concern the immediate adjacent hotspots.

- - same comment for the sentence lines 334-335 and for the sentence in the abstract, they do not see an effect of an insertion located just outside the cluster of hotspots but it is difficult to draw a general conclusions from one observation.

We agree with the reviewer, and we modified the manuscript to emphasize the limitation of our assay.

- They obtain a new allele *msh2* by performing CrispR Cas9 in the Ler background. How do they validate the allele ? have they looked at meiotic spreads ? pollen viability? Transcription ?

Mutation in the *MSH2* gene does not cause any visible phenotype that could be detected at the level of meiosis or fertility (Blackwell et al. EMBO J 2020, doi: 10.15252/embj.2020104858). Instead, we verified the mutant by RT-PCR amplification of the *MSH2* transcript from wild type and mutant plants. While the product of correct size was generated for the wild-type Ler plants, only the shortened product was amplified for the *msh2-6* mutant. A control reaction using RNA as template yielded no product confirming that the amplification was from mRNA and not from the genomic DNA. We cloned the RT-PCR products from the mutant and Sanger sequenced a few clones. They showed that the transcripts generated in *msh2-6* contain a deletion that produces a premature stop codon. As the mutation is located in 1/4 of the protein-coding sequence, it can therefore be presumed that *msh2-6* is a null mutant. We included this result in the revised manuscript as a Supplementary Fig. 9.

Reviewer #4 (Remarks to the Author):

DNA polymorphism effect on hotspot activity

The manuscript from the Ziolkowski lab uses a powerful seed-typing method to identify and fine-map crossovers in two crossover hotspot intervals. Using this method they undertake a considerable amount of work investigating the effect of polymorphism on CO rates in these intervals. Within the ChP and BT intervals CO rates were highest in regions containing fewer SNPs, but total COs were increased in the ChP interval in hybrids compared to pure lines. In contrast there was no change in COs in the BT interval in hybrids compared to pure lines. These results suggest an intriguing possibility that SNPs differentially affect CO designation at different scales – with SNPs stimulating COs at the (10s of) kilobase scale, but with COs being placed in small regions of homozygosity within these broader regions of high SNP density. Unfortunately, the experiments do not quite go far enough to convince me of this possibility.

We are glad that the reviewer appreciated the amount of work we put into preparation of this paper. We hope that the revised version of the manuscript, which includes the new way of presenting the results and new data, will be satisfactory.

The authors also showed that deletions within hotspots decreased their activity, but no compensatory increase was observed in neighbouring hotspots. In addition, large structural polymorphisms adjacent to the hotspots did not affect CO rates.

There is a lot to commend in this paper and it represents a considerable amount of work. However, I do have several issues. The first major issue is that in figure 1f and all similar representations, CO rates are not normalised by window length. This can be very misleading, for example if CO rates were uniform across the whole interval, regions of lower SNP density would artificially appear to have more COs because the COs occurring in a larger window (i.e. distance between adjacent SNPs) are being summed. The y-axis value would be proportional to the interval between adjacent SNPs. To get around this issue the y-axis value in 1f (and all similar plots) should be given in cM/Mb – this may require splitting the interval up into windows of similar size (as in 5d).

We thank the reviewer for this comment. Indeed, as the *ChP* and *BT* intervals are relatively long, using cM/Mb as a Y-axis scale was problematic due to some short sections between adjacent SNPs resulted in very noisy plots. The reviewer's suggestion stimulated us to find a satisfying solution to this problem. In the revised manuscript, we combined SNPs separated by less than 100 bp (for the *ChP* interval) or less than 400 bp (for the *BT* interval). This allowed us to present the data in a conventional way (cM/kb) with relatively little loss of resolution.

The representation in 5d is also problematic, as all COs are being recorded at the mid-point of the two flanking SNPs which gives an artificially high sense of the CO resolution. Again, for Figure 5d the y-axis units should be cM/Mb, with a single value per window given.

We agree with the reviewer that the way of presenting this data may be misleading. Therefore, we modified the plot on this figure (actual Fig. 6d) to show cM/Mb values for 13 defined segments of the *BT* interval. Polymorphism density has been represented in a form of bar plot corresponding to the CO plot.

My second major issue is one of over interpretation. "General" rules are stated, while the authors have only looked at two intervals and these two intervals show different results. The authors propose a plausible explanation for this, based on redistribution of COs due to differences in relative SNP density, however they have not performed the required experiments to convince me of this explanation.

We appreciate reviewer's comment in this aspect. The manuscript has been extensively rewritten to better explain some of our observations and to eliminate any overstatements from our conclusions. Importantly, we added new data supporting our hypothesis. More details on this will be indicated in a point-by-point response.

Undertaking CO analysis in R² BT lines and in R² BTmsh2 and R² ChPmsh2 lines would (if the results are in line with the hypothesis) make me much more convinced of the proposed explanation.

Unfortunately, we were not able to repeat the R² experiment for the *BT* interval. Constructing the R² lines requires an extremely long time (five generations which is at least 1.5 years in *Arabidopsis*). However, we present the results of the R² Col-*ChP* × *Ler msh2* experiment proposed by the reviewer (Figure 8e), which show that the effect depends on the presence of functional MSH2. Furthermore, we have included a new set of interesting data for the R² Col-*ChP* line in a cross with *Ler*, both in wild type and *msh2*; in this case the *ChP* interval is the only chromosome fragment that is homozygous while the remainder of the genome is heterozygous. We describe this result in more

detail below. Due to the high level of similarity between the observations for *ChP* and the observations presented earlier for *420* (Blackwell et al., 2020 doi: 10.15252/embj.2020104858), as well as the consistent results for the *BT* interval (although without the R^2 experiment), we believe that our research hypothesis has sufficiently solid experimental foundations to be presented in this paper.

While I have some issues with the manuscript, the question being addressed is an important one for the community and the field is full of apparently contradictory findings both on the influence of SNPs on recombination and the role of MSH2. There is much in this manuscript that helps shed light on these factors and how they interact to shape CO distributions.

We are pleased that the reviewer appreciates the work.

Additional specific points are given below.

Line 42: is repaired by -> are repaired as

We believe that in this sentence “a small fraction” is the subject thus the form of the verb is correct.

Line 73: I think there is still much to be learned regarding the relationship between SNPs and COs – and some of this could be brought out more in the intro. The lack of a juxtaposition effect in *msh2* is supportive of SNPs having an MSH2 mediated pro-crossover effect. On the other hand, it has recently been reported by the Mercier group (Lian et al 2022, Nat. Comm.) that the megabase-scale crossover landscape is largely independent of sequence divergence with recombination distributions in *Arabidopsis* pure lines being very similar to those of Intra-specific hybrids (e.g. highest recombination rates in pericentromeric regions in pure lines). This supports the notion that much of the association of CO rates and SNPs is due to mutagenic effects of recombination rather than a pro-crossover influence of SNPs.

We thank the reviewer for this remark, with which we fully agree. In the introduction, we added an excerpt about the latest discoveries from the Mercier group. The juxtaposition of these seemingly contradictory results from various research groups allows to emphasize the complexity of the interaction between polymorphism and meiotic recombination, which better justifies the further need to study this problem.

Line 144: This seems excessively high? Or is this combined depth for all recombinants rather than depth per amplicon?

The value given applies to each library (thus, to each recombinant; consisting of three or five amplicons for *ChP* and *BT*, respectively) and not to the entire pool. Indeed, the sequencing depth we use is very high. This is due to the sequencing technology used (HiSeq X-10) and the fact that we sequence a relatively short region (26 kb for *ChP* and 49 kb for *BT*). On the other hand, recombinant selection, high-molecular weight DNA extraction, interval amplification by LR-PCR and library construction are demanding and time-consuming, so by increasing the depth of sequencing, we reduced the risk of losing some data.

Line 148 – Figure 1f: Y-axis value should be normalised by window length e.g. cM/Mb, otherwise I cannot tell if regions with a high y-axis value truly have higher recombination rates, or whether it's just a case of more recombination events being detected because the regions are larger (i.e. greater distance between adjacent SNPs).

As already mentioned, we agree with this remark and all plots in the manuscript have been normalized to the SNP-SNP window size and remade using the cM/Mb scale.

In order to really see the recombination rate across the region in a comparable way, the y-axis needs to be converted to cM/Mb. For very small intervals between adjacent SNPs this will likely generate a very noisy plot, so it may make more sense to plot the cM/Mb value across the region using a set window length, rather than intervals of variable length determined by the spacing of adjacent SNPs.

This suggestion was very helpful as we were concerned that there would be high local recombination values when using cM/Mb scale. Applying an approach in which we combined the shorter SNP-to-SNP sections (<100 bp for *ChP* and <400 bp for *BT*), we managed to maintain good resolution while significantly reducing the noise.

Line 179: Perhaps I am mis-understanding something here, but I don't follow the logic. Why would the presence of the fluorescent proteins affect recombination?

For dsRED the T-DNA is still present so presumably the transgene is still transcribed even if it doesn't result in a functional mRNA? Given transcriptionally active genomic regions have higher recombination rates mightn't the (non-functional) T-DNA still affect COs? How does the recombination rate in a Col-ChP x Ler measured using the

fluorescent reporters correspond to the CO rate determined in other published genome wide recombination analyses? E.g. Rowan et al. Wouldn't this also answer the question?

The Col-*ChP* line differs from the Col line only in the presence of reporter cassettes, which are necessary for measuring recombination. However, from a genetic point of view, these cassettes should be considered as DNA insertions in the Col line. Since the *ChP* interval is very short, the presence of these two insertions can influence the frequency of recombination. Therefore, we wanted to check if recombination within *ChP* changes after crossing Col-*ChP* with a line that also had those insertions (in this experiment, the presence of the insertion was the main factor not the expression of the reporter genes). However, it was necessary to disable the functionality of both cassettes so that the segregation of reporters in the cross could still be observed. This is what we have tried to do with CRISPR targeting both cassettes. We managed to achieve it for the dsRed cassette, but not for the eGFP, which was virtually deleted. Thus, we consider this experiment as partially successful, as we could conclude about the effect of the dsRed cassette, but not the eGFP cassette. We agree with the reviewer that it was not clearly explained, we hope that the revised version will be easier to understand.

We also thank for the suggestion regarding the comparison of CO rate between Col-*ChP* × *Ler* and Col × *Ler* according to Rowan et al. that we have included in this revised paper.

Or is this more about whether plants are hemizygous for the T-DNA insertion – the idea being that there is structural polymorphism between the two homologs in Col-*ChP* × Col, but not in Col-*ChP* × Col- Ψ ChP? If the GFP cassette is completely deleted in Col- Ψ ChP doesn't this mean that there is still structural polymorphism in both Col-*ChP* × Col and Col-*ChP* × Col- Ψ ChP? In which case, my interpretation would be that it is not possible to determine whether the T-DNA influences CO rates. Am I missing something?

We agree at this point that there is still a structural polymorphism due to the presence of the eGFP cassette. However, in the Col- Ψ ChP line, we eliminated the effect of the dsRed insertion. This was emphasized in the revised manuscript.

The nature of the CRISPR mutations is also very unclear. First in the text it states the authors generated a “pseudo-reporter line in which the reporter cassettes were preserved but produced nonfunctional fluorescent proteins”. Later in the text it says the GFP cassette was “largely removed” and in the Figure S4 legend it states the GFP cassette was completely removed. In the legend to Figure 2 it says GFP has a missense mutation. Which is it?

For dsRED the main text and Figure S4 state that there is a frameshift mutation, while in the Figure 2 legend it says dsRED has a missense mutation. Again, which is it?

We apologize for this inconsistency. As we already explained, missense mutation was obtained for the dsRed cassette, and an almost complete deletion was generated for the eGFP cassette. We corrected both the manuscript text and the corresponding figure captions accordingly.

Figure S4 needs more detail, what sequences are being aligned here? Are the chromatograms being aligned with wild-type genomic DNA for GFP and the T-DNA sequence for RFP? This needs better labelling.

The figure has been remade; this is now Supplementary Fig. 3.

Line 192: Was this tested statistically? Needs to be included in text, figure or table if claims of significance made. Perhaps a supplementary table with pair-wise significance tests?

The results of the statistical test were shown in Figure 3d. In the new version of the manuscript, we also indicated this fact in the main text so that it would be easier to find by the readers.

Line 240: “Similar to ChP, BT inbreds showed no change from the wild type” – Presumably the BT inbreds referred to here are *msh2* mutants? Would be good to make this explicit.

We thank for this comment. The reviewer is right and we now clearly stated this in the manuscript.

Line 255: This general statement is not supported by the data. It is only true for the ChP hotspot, not for the BT hotspot (Figure 4a-b). This should be modified to refer only to the ChP hotspot. Or is this referring to finer scale hotspots within the ChP and BT intervals??

Indeed, this subheading refers to the crossover distribution within the interval, which is the finer scale. As this subheading seems too ambiguous, we have made it more specific to read: “MSH2 stimulates crossovers in hotspots surrounded by SNP-rich regions”.

If this is the case, I'm not sure how it is possible to show that SNPs stimulate COs, given that it is impossible to work out the distribution of CO events within an interval in the absence of any polymorphism.

This is shown by our R²-2 lines (4 independent lines), which differ from the Col-*ChP* x Col inbreds by the presence of only 40 - 44 SNPs located exclusively within one hotspot (*Coco*) (Fig. 8b-d). All these lines show a higher *ChP* CO rate than inbreds (although in one case the increase is not statistically significant). We do not know how COs are distributed in these lines, but in R²-1 lines, in which the entire *ChP* interval is heterozygous, the *ChP* CO rate is even higher. Moreover, in the R²-1 and R²-2 siblings in the *msh2* background the effect disappears, confirming that it is dependent on the detection of SNPs by MSH2.

It is possible to make the argument that MSH2 promotes COs in a SNP density dependent manner at the individual hot-spot scale (Fig 4e and Figure 5e). It does not follow however that SNPs promote COs at the hotspot scale, there is no apparent correlation between SNP density and CO rate in figures 4C and 5a/5d, though this has not been looked at explicitly. To do this it would be best to split the intervals up into windows of similar size (as has been done for BT in figure 5d) and then do a scatter plot comparing SNP density with CO rate measured in cM/Mb. It would then be possible to test if there is any correlation between SNP density and CO rate i.e. to see if "SNPs stimulate COs at the hotspot scale".

It seems to us that the issue raised here by the reviewer is the result of a lack of precision on our part. There is no doubt that SNP polymorphism plays a relatively small role, if any, in determining recombination hotspots. Both the work of Henderson's and Mercier's groups show that the location of hotspots is primarily due to epigenetic factors, especially the open chromatin structure. Our work clearly shows that the hotspot center is basically devoid of polymorphisms, similar conclusions can be drawn by studying CO topology in the pollen-typing data (e.g., Serra et al. 2018, doi: 10.1371/journal.pgen.1007843; Choi et al. 2016, doi:10.1371/journal.pgen.1006179). Moreover, we show that the location of hotspots in the *msh2* mutant, which should be insensitive to polymorphisms, is practically the same as in the wild type (Fig. 5c and 6a). From this perspective, it seems pointless to investigate the correlation between SNP density and CO rate along the intervals, as we do not believe there is any correlation.

However, changes in the recombination activity of individual hotspots in *msh2* relative to the WT show that hotspots surrounded by multiple SNPs are more active in the WT than hotspots in non-polymorphic regions. The results in this respect are consistent for both *ChP* and *BT* (Fig. 5e and 6e). As this is dependent on MSH2 (in *msh2* the effect disappears and in some heterozygosity contexts seems to be even opposite, see Fig. 8e), we believe that our conclusion about MSH2 stimulating crossovers in hotspots based on locally occurring interhomolog polymorphisms is supported by our data for both intervals.

I also do not like the representation in figure 5d which gives an artificially high sense of the resolution of CO mapping. The narrow widths of the blue and red peaks imply a very narrow distribution of CO positions where in reality the resolution of CO position is limited by SNP density. For example, the large peak in interval 4 is high because all COs in a 5kb region between adjacent SNPs are being recorded at the midpoint between the two SNPs. Also what does the metric CO percent correspond to? Is this the proportion of all BT COs that occur in this window? This is problematic given that the windows are of different sizes. Normalising by window length (e.g. cM/Mb). Also, a single CO value (in cM/Mb) and SNP density value (polymorphisms/kb) should be used for each window.

We agree with the reviewer that the presentation of the data in the figure (now Figure 6d) can be misleading. Therefore, we changed this figure in the way suggested by the reviewer.

Line 291: This ambiguous, it sounds like you are looking at recombination events that occurred in the 4.5 Mb genomic region in-between BT and ChP.

We apologize for this ambiguity, this has been corrected.

Line 311++: Again, I do not agree with this statement. For one interval (ChP) there was higher recombination in a hybrid compared to pure lines, for a second interval (BT) there was no change in recombination in a hybrid compared to pure lines. It is impossible to make a "general" statement when two intervals were analysed and they both responded differently to polymorphism and *msh2* mutation (Figure 4a / 4b).

In our opinion, the fact that *BT* CO rate in hybrids decreases in relation to inbreds, and in *ChP* it is the opposite, cannot be interpreted as an argument saying anything about the effect of polymorphism on CO. This is because chromosomal CO landscape is shaped by the effects of different genetic backgrounds (Col vs. Col/*Ler*) as well as by CO interference effects that lead to a redistribution of CO while keeping the same CO numbers. The conclusions were based on the comparison of crossover topology changes in *msh2* vs. wild type: for both intervals we observed that polymorphic hotspots/regions are more active in WT than in *msh2* (Figs 5e and 6e).

However, we agree with the reviewer that the effect is MSH2 dependent, therefore we rephrased the aforementioned text excerpt.

One possibility is that in a hybrid the effect of higher SNP density in pericentromeric regions (e.g. *ChP*) dominates any potential stimulatory effect of SNPs in chromosome arms. Indeed, a wealth of recent papers show that ZMM crossover numbers are set by HEI10 dosage, so SNP density could at most be expected to cause a redistribution of COs toward regions of higher SNP density rather than stimulate additional COs. In fact, as *Ler* has a weaker HEI10 allele than *Col*, hybrids have less COs than pure *Col* (Lian et al 2022, Nat. Comm.).

We agree that the observed differences between *BT* and *ChP* are a consequence of CO redistribution, while the total number of crossover events remains unchanged in the hybrid versus inbred (or, as the reviewer rightly noted, even reduced in hybrid).

Line 365: I do not find this surprising given the same regions that show highest levels of polymorphism between accessions also show the highest levels of recombination in pure lines (Lian et al 2022, Nat. Comm.). This suggests there is some other factor (i.e. not inter-homolog polymorphisms) is contributing to chromosomal scale CO distributions in Arabidopsis.

The effect described in the work of Lian et al. 2022 is about a genome-wide comparison of inbreds and hybrids. However, the situation is different when only a fragment of the chromosome is heterozygous, while the rest of the chromosome is homozygous (or vice versa): then in the wild-type Arabidopsis there is a redistribution of CO into heterozygous regions (the effect was tested at three different intervals; *420*, *12f* and *CEN3*; Ziolkowski et al. 2015). We expected that phenomenon will be reflected to some extent in the kb scale, with a short distance-distribution of COs from the less to more polymorphic sites. Therefore, it was surprising to us that the site of the CO event is practically polymorphism-free. In the new version of the manuscript, we rephrased this sentence to be more specific.

Line 374: Check wording here.

Corrected.

Line 377: Sorry I don't follow this reasoning. Was there any statistical test done comparing CO rates in *msh2* pure lines and hybrids? These don't appear on the Figure referenced. In the absence of *msh2* there appears to be no change in COs between pure lines and hybrids for both *ChP* and *BT*? What is the relevance of the HEI10-OE lines?

As this part of the text was not crucial to the key conclusion in this paragraph (saying that "MSH2 complexes actively stimulate CO repair in polymorphic hotspots"), we have removed this fragment.

To be fully convinced that SNPs generally stimulate COs (and that it is not largely a position effect, or an effect restricted to the peri-centromeric regions) I would need to see data for R² lines for the *BT* interval. If SNPs do stimulate COs then it would be expected that in contrast to a *Col/Ler* hybrid (where SNP density in *BT* is lower than pericentromeric regions), CO rates would increase in a *BT* R² line (SNP density in the *BT* region being higher than in pericentromeric regions).

Although for time constrains we are unable to construct the R² line for the *BT* interval (this would require five generations which is at least 1.5 years for Arabidopsis), we believe that such a proof is not necessary. First, our analyzes of the CO distribution in *BT* in the wild-type and *msh2* mutant backgrounds show very clearly the correlation between the change in CO activity and the level of polymorphism (Fig. 6). This correlation is statistically significant. Second, we added new results for the *ChP* interval: by crossing the R² line with *Ler* we obtained plants in which *ChP* was homozygous while the remainder of the genome was heterozygous (Fig. 8b). *ChP* CO frequency was lower in those plants than in the hybrids. When we repeated this experiment in the *msh2* background, the effect disappeared. On this basis, we prepared a new figure 8e where we included those new results and presented the CO frequency in different heterozygous contexts, similar to our earlier work (Blackwell et al., 2020 doi: 10.15252/emj.2020104858). For the convenience of the reviewer, we present these two datasets side by side below, with the same order of lines of analogous heterozygosity pattern; this comparison was also included in a new Supplementary Fig. 11.

As the reviewer will notice, the pattern is very remarkable. The differences between 420 and ChP can be observed for the comparison of inbreds and hybrids, both in WT and in *msh2*. This may be the result of a different genetic background (Col/Ct for 420 and Col/Ler for ChP) or chromosomal location (subtelomeric and pericentromeric, respectively). However, the effects for lines with different heterozygosity pattern, which reflects the response to local polymorphism, are virtually identical. This indicates that the effects we observe for ChP is the same as the juxtaposition effect described before, albeit with much longer intervals. It is worth noting that the juxtaposition effect in WT and *msh2* was described for the subtelomeric interval 420, and for WT also for the subtelomeric interval *12f* (chromosome 2) and pericentromeric *CEN3* (chromosome 3) (Ziolkowski et al., doi:10.7554/eLife.03708).

To be convinced that the effect is MSH2 dependent (and not resulting from a SNP independent CO redistribution in *msh2* mutants) I would also need to see results for a BT and ChP R2 line in an *msh2* background. If the authors' hypothesis is correct, then both ChP and BT R2 *msh2* lines should show no change in CO compared to inbred wt.

As requested by the reviewer, we present the results for ChP R² in the *msh2* background (Figs. 8d and 8e). As expected, they show no significant difference from inbreds. Due to time constraints, we did not conduct similar experiments for BT, but given that the main objection of the reviewer concerned the chromosomal localization, we believe that the published data for the 420 interval should provide sufficient support for the proposed conclusion. CO frequency in line "Heterozygous in 420 and homozygous elsewhere" is significantly higher than in inbreds (26.2 cM vs. 18.2 cM, $p = 5.5 \times 10^{-15}$), however in the *msh2* background the trend is reversed and recombination is even slightly lower than in inbreds (20.4 cM vs. 23.2 cM, $p = 1.9 \times 10^{-3}$). This suggests that MSH2 stimulates crossover recombination in polymorphic regions regardless of the chromosomal location.

Reviewers' Comments:

Reviewer #1:

Remarks to the Author:

The authors have satisfactorily addressed all my concerns.

Reviewer #2:

Remarks to the Author:

I believe the authors addressed the reviewers' queries in a satisfactory manner and I have no further comments.

Reviewer #3:

Remarks to the Author:

The authors have considered my comments and modified the manuscript accordingly. I have no further comments

Reviewer #4:

Remarks to the Author:

The revised manuscript is much improved. Particularly, the toning down of conclusions and the alteration to the figures to present CO rates in cM/Mb. Supplementary figure 5 should also be updated to y-axis in cM/Mb.

Minor point

Line 275: This is not a general feature, in one of the two intervals investigated (BT) CO levels are lower when MSH2 is present. This heading needs clarification.

Dear reviewers,

Our responses to your comments are highlighted in blue type.

Reviewer #1 (Remarks to the Author):

The authors have satisfactorily addressed all my concerns.

Reviewer #2 (Remarks to the Author):

I believe the authors addressed the reviewers' queries in a satisfactory manner and I have no further comments.

Reviewer #3 (Remarks to the Author):

The authors have considered my comments and modified the manuscript accordingly. I have no further comments

Reviewer #4 (Remarks to the Author):

The revised manuscript is much improved. Particularly, the toning down of conclusions and the alteration to the figures to present CO rates in cM/Mb. Supplementary figure 5 should also be updated to y-axis in cM/Mb.

We are pleased that all reviewers are satisfied with the changes we have made to our revised manuscript. With reference to Reviewer #4's comment, we have modified Supplementary Figure 5 as requested.

Minor point

Line 275: This is not a general feature, in one of the two intervals investigated (BT) CO levels are lower when MSH2 is present. This heading needs clarification.

We previously explained that the title refers to crossover distribution at the finer scale and not between different chromosomal regions. Therefore, the feature seems to be general, as the same observations were made both for *ChP* and *BT* intervals at this scale. For reference, below is a relevant excerpt from a previous review:

<<<

Line 255: This general statement is not supported by the data. It is only true for the ChP hotspot, not for the BT hotspot (Figure 4a-b). This should be modified to refer only to the ChP hotspot. Or is this referring to finer scale hotspots within the ChP and BT intervals??

Indeed, this subheading refers to the crossover distribution within the interval, which is the finer scale. As this subheading seems too ambiguous, we have made it more specific to read: "MSH2 stimulates crossovers in hotspots surrounded by SNP-rich regions".

>>>

However, to simplify and generalize this header, we finally changed it to read: "MSH2 stimulates crossovers in hotspots in response to local polymorphism". We are confident that this title will be accepted by the reviewer.